# 2D quasi-layered material with domino structure

Haihui Lan[1,2,6], Luyang Wang[3,6], Runze He[1,6], Shuyi Huang[3,6], Jinqiu Yu[3], Jinming Guo[4], Jingrui Luo[3], Yiling Li[3], Jinyang Zhang[4], Jiaxin Lin[5], Shunping Zhang [5], Mengqi Zeng [3] ✉ & Lei Fu [1,3] ✉

Interlayer coupling strength dichotomizes two-dimensional (2D) materials into layered and non-layered types. Traditionally, they can be regarded as atomic layers intrinsically linked via van der Waals (vdW) forces or covalent bonds, oriented orthogonally to their growth plane. In our work, we report a material system that differentiates from layered and non-layered materials, termed quasi-layered domino-structured (QLDS) materials, effectively bridging the gap between these two typical categories. Considering the skewed structure, the force orthogonal to the 2D QLDS-GaTe growth plane constitutes a synergistic blend of vdW forces and covalent bonds, with neither of them being perpendicular to the 2D growth plane. This unique amalgamation results in a force that surpasses that in layered materials, yet is weaker than that in non-layered materials. Therefore, the lattice constant contraction along this unique orientation can be as much as 7.7%, tantalizingly close to the theoretical prediction of 10.8%. Meanwhile, this feature endows remarkable anisotropy, second harmonic generation enhancement with a staggering susceptibility of 394.3 pm V$^{-1}$. These findings endow further applications arranged in nonlinear optics, sensors, and catalysis.

Two-dimensional (2D) materials have emerged as a prominent research focus owing to their extraordinary attributes[1–7], with wide application in a diverse array of fields[8–18]. In the past decades, 2D materials manifest two distinct forms: layered and non-layered[19], based on the strength of the interlayer coupling effect. They can be regarded as atomically thin layers that are interconnected either by vdW forces or covalent bonds. The atomic interactions transpire perpendicularly to the plane corresponding to the material's bi-dimensional growth, thereby yielding a distinct structural configuration[6]. This orthogonal relationship is primarily attributed to the propensity of atoms to adopt a tightly packed arrangement, which serves to minimize entropy and bolster structural stability[6,14,20]. Consequently, such configurations thwart any potential deformation or disintegration of the material[21],

posing a formidable challenge in unveiling the underlying structure beyond these interactions.

In a groundbreaking advance, we report a hitherto unknown material concept that totally differentiates from traditional layered and non-layered materials. This exceptional material displays special interlayer interactions that consist of a synergistic amalgamation of vdW forces and covalent bonds, with neither of them being perpendicular to the 2D growth plane, considered as the domino tiles. This unique amalgamation results in a force that surpasses that in layered materials yet is weaker than that in non-layered materials. We have coined this captivating structural configuration as a 2D quasi-layered material with a domino structure, signifying its distinct position within the landscape of 2D materials.

[1]The Institute for Advanced Studies, Wuhan University, 430072 Wuhan, China. [2]Department of Chemistry, Massachusetts Institute of Technology, Cambridge, MA 02139, USA. [3]College of Chemistry and Molecular Sciences, Wuhan University, 430072 Wuhan, China. [4]Key Laboratory of Green Preparation and Application for Functional Materials, Ministry of Education, School of Materials Science and Engineering, Hubei University, 430062 Wuhan, China. [5]School of Physics and Technology, Wuhan University, 430072 Wuhan, China. [6]These authors contributed equally: Haihui Lan, Luyang Wang, Runze He, Shuyi Huang. ✉e-mail: zengmq_lan@whu.edu.cn; leifu@whu.edu.cn

2D quasi-layered materials with domino structures display unique properties distinct from those of layered and non-layered materials. As a representative, we proposed a 2D quasi-layered domino-structured gallium telluride (QLDS-GaTe) that differs from the traditionally layered configuration. QLDS-GaTe reveals a distinctively skewed growth structure, diverging from the substrate orientation by an approximate angle of 25°. To fortify this intriguing conformation, a pronounced enhancement in interlayer coupling becomes discernible. Remarkably, the lattice constant contraction along this unique orientation can be as much as 7.7%, tantalizingly close to the theoretical prediction of 10.8%. This unique material demonstrates remarkable anisotropy coupled with a pronounced second harmonic generation (SHG) enhancement effect, its polarization tensor achieving a staggering 394.3 pm V$^{-1}$. This intriguing property holds potential for future exploration within the realm of nonlinear optics.

## Results

### Growth strategy and stability of 2D QLDS-GaTe single crystal

In crystallography, atoms in crystal configurations are predisposed to a highly dense arrangement, which can mitigate entropy and augment structural stability. This effect becomes particularly evident in layered materials propagated on a traditionally vdW substrate. In such systems, interatomic forces generally manifest a symmetric distribution within a 2D plane parallel to the substrate (Fig. 1a), thereby leading to a random distribution of growth orientations on the substrate (Supplementary Fig. 1). The traditional growth mode of 2D materials with the best stable layered configuration is represented by planar growth mode.

To break the uniformity of interaction strength in the aforementioned 2D plane, the interaction-strength-modulation growth (ISMG) strategy has been proposed. This strategy aims to disrupt the uniform interaction and augment the interaction strength between the substrate and the crystal in a direction orthogonal to the plane, leading to a phenomenon known as skewed growth (Fig. 1b). A liquid metal was chosen as the growth substrate owing to its inherently flat surface, significant capacity to accommodate additional elements and well-matched interactions with skewed growth mode[22–25]. The crystalline attribute, as evidenced by the temperature-dependent X-ray diffraction (XRD) diagram (Supplementary Fig. 2), suggests that the Ga substrate becomes a highly interactive substrate. This substrate facilitates significant charge interactions with the material nucleating on its surface and disrupts the conventional planar growth mode of the materials (Supplementary Fig. 3). Simultaneously, the as-nucleated material induces the precursor atoms dissolved in liquid metal to accumulate on their interface since the layering effect[25–27], further promoting the crystalline process (Supplementary Fig. 4). Therefore, ISMG strategy with liquid metal can reduce the formation energy across an expansive chemical potential window tuned by temperatures (Supplementary Figs. 5–7) and facilitates an inclined crystal growth with a well-aligned growth orientation (Supplementary Fig. 8). Additionally, by extending the annealing duration, the concentration of Te dissolved in the liquid metal progressively increases, which in turn affects the thickness of the material (Supplementary Figs. 9 and 10). The ISMG strategy exhibits universality. Through this approach, we have also synthesized indium telluride materials with a domino structure (Supplementary Fig. 11). This discovery not only broadens the applicability of the method but also lays a foundation for unveiling more 2D QLDS materials.

The skewed growth mechanism engenders the formation of domino-structured material characterized by a distinct interaction paradigm. In this model, this exceptional material exhibits an acute angle of inclination between its layers and the horizontal plane, considered as the domino tiles. It displays special interlayer interactions that consist of a synergistic amalgamation of vdW forces and covalent bonds. This unique amalgamation results in a force that surpasses that

in layered materials yet falls short of that in non-layered materials (Fig. 1c). Consequently, we categorize this kind of material as a quasi-layered substance. As an exemplar, the 2D QLDS-GaTe unveils a uniquely skewed growth structure, diverging from the substrate orientation by approximately 25°. This deviation indicates a cooperative interplay of van der Waals forces and covalent bonds, neither of which aligns perpendicularly to the 2D growth plane. Significantly, the impact of a densely packed arrangement persists in this class of quasi-layered materials. To ensure structural stability, an evident enhancement in interlayer coupling transpires, contributing to entropy minimization and structural fortification. Therefore, the interlayer interaction strength exceeds that of the vdW forces, as evidenced by the existence of high electron density between layers shown in the differential charge density plot (Fig. 1d).

Furthermore, to corroborate the structural stability of 2D QLDS-GaTe at ambient conditions, we conducted calculations of the phonon dispersion curves at a temperature of 300 K, as shown in Fig. 1e. The resultant phonon spectrum for the 2D QLDS-GaTe demonstrated that all phonon branches remained positive across the entirety of the Brillouin zone, thereby affirming its structural stability at room temperature. Molecular dynamics simulations performed at a temperature of 300 K manifested no discernible energy drift or structural dissociation (Fig. 1f and Supplementary Fig. 12), highlighting the substantial stability of 2D QLDS-GaTe. Collectively, these findings robustly substantiate that the 2D QLDS-GaTe retains its post-synthesis stability on a liquid metal substrate.

### Structure characterization of 2D QLDS-GaTe single crystal

As delineated in Fig. 2a, the 2D domino-structured material needs the amplification of interlayer forces to ensure structural stability, a process that concurrently enables entropy minimization and structural reinforcement. Drawing an analogy to domino tiles, wider inter-tile spacing is required when larger tiles are inclined at an identical angle, while their smaller counterparts necessitate narrower spacing. The correlation between the lattice constant and thickness is depicted in Fig. 2b. On approaching the 2D limit (-1 nm), there is a marked contraction in the lattice constant in the stacking direction, with the maximum shrinkage extending to as much as 10.8%. This emphatically underscores the remarkable potential inherent in the ISMG strategy. The approach facilitates the synthesis of materials that can approach the 2D limit of 1.2 nm (Supplementary Fig. 13). Moreover, it yields 2D QLDS-GaTe specimens of diverse thicknesses, thereby presenting a compelling platform for investigating interlayer coupling phenomena.

To scrutinize the enhancement of interlayer coupling effects during the transition to two-dimensionality in QLDS-GaTe, samples of disparate thicknesses were examined. All samples demonstrated a single set of diffraction structures in their Fast Fourier Transform (FFT) patterns (Fig. 2c, f), confirming the robust crystallinity across QLDS-GaTe samples of diverse thicknesses synthesized via the ISMG strategy. These FFT patterns align closely with the calculated simulated FFT diffraction structures (Supplementary Fig. 14), corroborating the synthesis of GaTe with a domino structure on the (−101) plane. Furthermore, using cross-sectional HADDF-STEM, we directly observed a slanted atomic arrangement in the material, offering additional validation that the material is indeed 2D QLDS-GaTe (Supplementary Figs. 15 and 16).

Notably, the interlayer (101) spacing in the high-resolution transmission electron microscopy (HRTEM) images of samples with various thicknesses were delineated (Fig. 2d, g), measuring 7.33 Å for the thin sample (-1 nm) and 8.06 Å for the thicker one (-10 nm). These results harmonize with the simulated HRTEM images of 1.2 nm and bulk samples (Fig. 2e, h), further affirming the domino structure of the GaTe (−101) plane. The lattice constant in the material undergoes a 7.7% contraction during the two-dimensionalization process, a value nearing the theoretical limit of 10.8%. Simultaneously, energy-dispersive X-

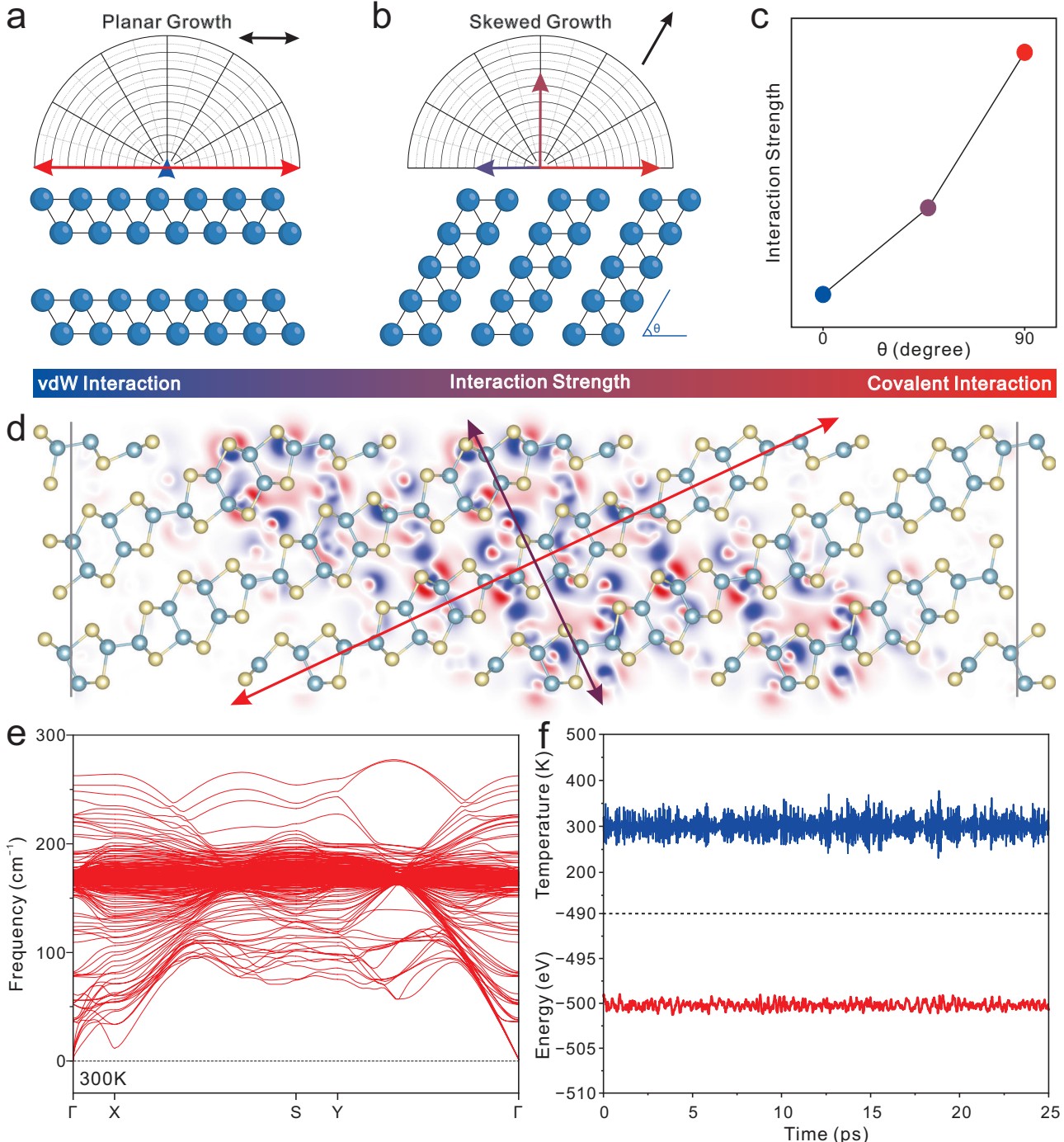

**Fig. 1 | Interactive-strength-modulation growth strategy and stability of 2D quasi-layered domino-structured gallium telluride (QLDS-GaTe) single crystal. a** Depiction of the interaction and growth orientation of layered GaTe on van der Waals (vdW) substrate. **b** Illustration of the interaction and growth orientation of QLDS-GaTe on the strong interactive substrate. **c** Scheme of skewed angle-related interaction strength perpendicular to the 2D growth plane. **d** Calculated contour maps of charge density difference for 2D QLDS-GaTe structure. Electron accumulation (depletion) is shown in red (blue) in accordance with the positive (negative) value below. Yellow atoms refer to Te atoms, and blue atoms refer to Ga atoms. **e** Calculated phonon dispersive curves for 2D QLDS-GaTe structure at a temperature of 300 K. **f** Temperature and total energy trajectories as derived from molecular dynamics simulations of the 2D QLDS-GaTe structure, conducted at 300 K.

ray spectroscopy (EDS) elemental mapping confirmed uniform and strong signals of Ga and Te elements throughout the crystal (Fig. 2i), highlighting the aptness of 2D QLDS-GaTe as a platform for probing interlayer coupling effects.

## Investigation of SHG in 2D QLDS-GaTe structure

SHG characterization serves as an effective method for identifying material symmetry in nonlinear optics[28]. The SHG investigation scheme applied to the 2D QLDS-GaTe structure elucidates its structural symmetry (Fig. 3a). A femtosecond laser with a steady output power of 1 mW was employed to incite the SHG signal. Remarkably, the 2D QLDS-GaTe structure exhibited a substantial nonlinear optical response within a range of 570–670 nm, corresponding to an excitation wavelength of 1140–1340 nm (Fig. 3b). Subsequent power-dependent investigations were carried out on the SHG intensity at a fixed wavelength. The spectrum presented in Fig. 3c attests to the

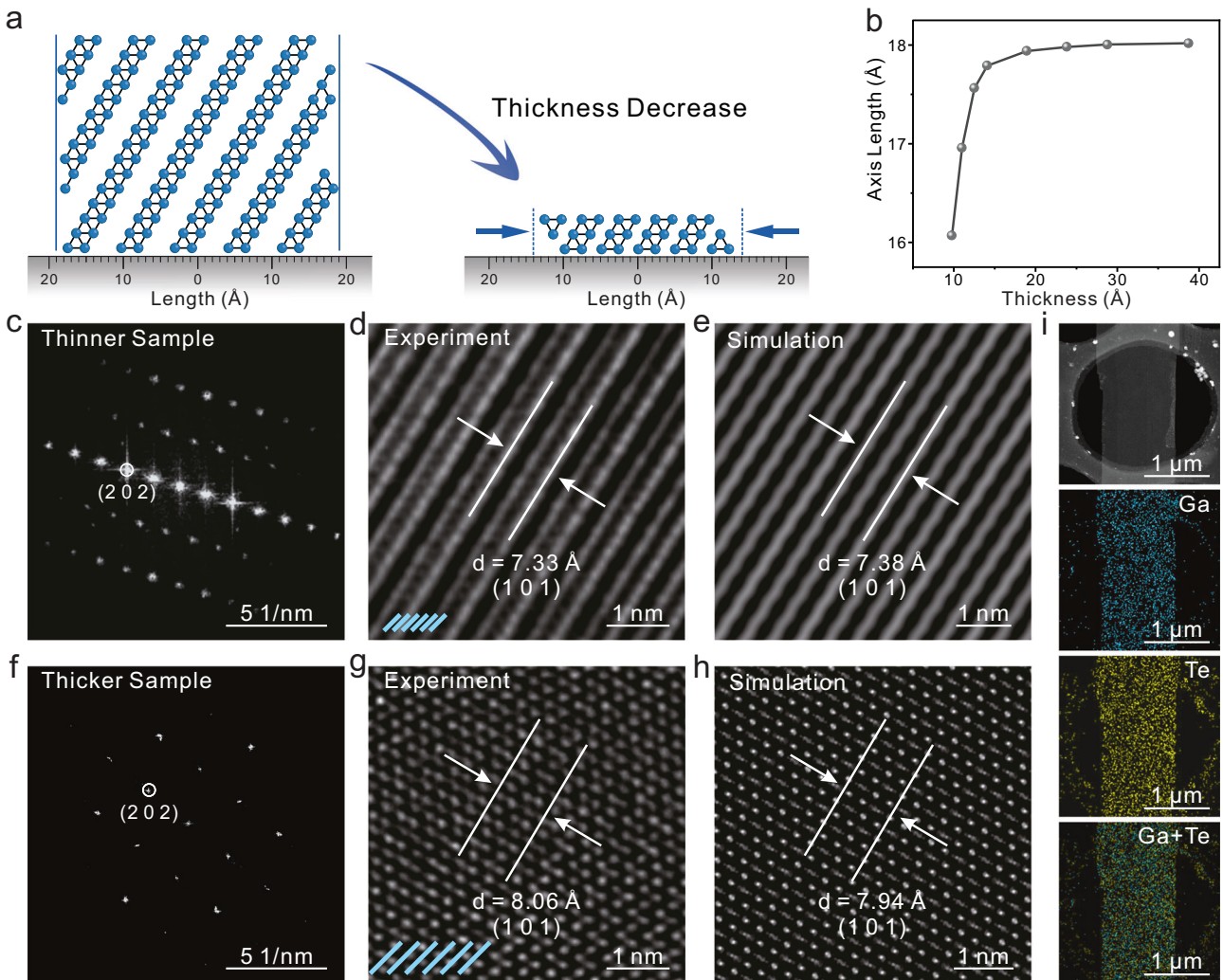

**Fig. 2 | Structural characterization of the 2D quasi-layered domino-structured gallium telluride (QLDS-GaTe) single crystal. a** Schematic depiction of the alterations in the lattice constant along the stacking direction following the transition to two-dimensionality in QLDS-GaTe. **b** Computed correlation of lattice constant in the stacking direction with respect to thickness. **c** Fast Fourier Transform (FFT) pattern derived from a relatively thinner sample of 2D QLDS-GaTe single crystal. **d**, **e** Experimental and corresponding simulated high-resolution transmission electron microscopy (HRTEM) images of the thinner 2D QLDS-GaTe single crystal. **f** FFT pattern derived from a comparatively thicker sample of 2D QLDS-GaTe single crystal. **g**, **h** Experimental and corresponding simulated HRTEM images of the thicker 2D QLDS-GaTe single crystal. **i** Low-magnification scanning transmission electron microscopy (STEM) image accompanied by corresponding EDS elemental mapping of Ga, Te, and the overlay of Ga and Te elements, observing from the planar perspective.

significant increase in SHG intensity with a tuning power range from 0.355 to 1.014 mW. The extracted power-dependent SHG intensity is displayed in Fig. 3d, revealing a double logarithmic relationship between the SHG intensity and laser power. The slope of the fitted line is 1.72, aligning with the theoretical value of 2[29], as predicted by the electric dipole approximation. Intriguingly, a marked increase in the SHG response signal is observed as the thickness of the material decreases. As shown in Fig. 3e, f, a sixfold increase in SHG response signal intensity is observed when the thickness is reduced to 4 nm from 20.5 nm. The calculated second-order nonlinear optical susceptibility is found to be 394.3 pm V$^{-1}$, considered an impressively substantial value, highlighting the superior nonlinear optical property of the 2D QLDS-GaTe single crystal.

We ascribe the anomalous intensification in the SHG to a restructuring in the band architecture (Supplementary Fig. 17), incited by alterations in the bonding mechanism of the material. Density functional theory (DFT) calculations reveal that the bonding mechanism of 2D QLDS-GaTe significantly differs from that of the bulk structure (Supplementary Fig. 18) due to the reconstruction of the surface

atoms of 2D QLDS-GaTe, thereby introducing new bonding states. Hence, we categorize the bonding arrangement in 2D QLDS-GaTe as an amalgamation of surface and inner bonding states. The inner bonding state, akin to the bulk structure, manifests virtually identical orbital hybridization and bond strength (Supplementary Figs. 19–22 and Supplementary Tables 1 and 2). On the contrary, the interactions of reconstructed surface atoms significantly deviate from the bulk structure (Supplementary Figs. 23 and 24 and Supplementary Tables 3 and 4). Taking Ga atoms as a case in point, they manifest a mixed state of $sp^2$ and $sp^3$ hybridization on the surface, while those in the bulk structure retain an exclusive $sp^3$ hybridization. Consequently, as the thickness decreases, the proportion of the surface charge state in the overall charge state gradually increases, leading to changes in the band structure. This is the underlying reason for the significant enhancement of the SHG signal as the thickness decreases.

## Discussion

In this work, we unveil a material concept named 2D quasi-layered material, filling the blank between layered and non-layered materials

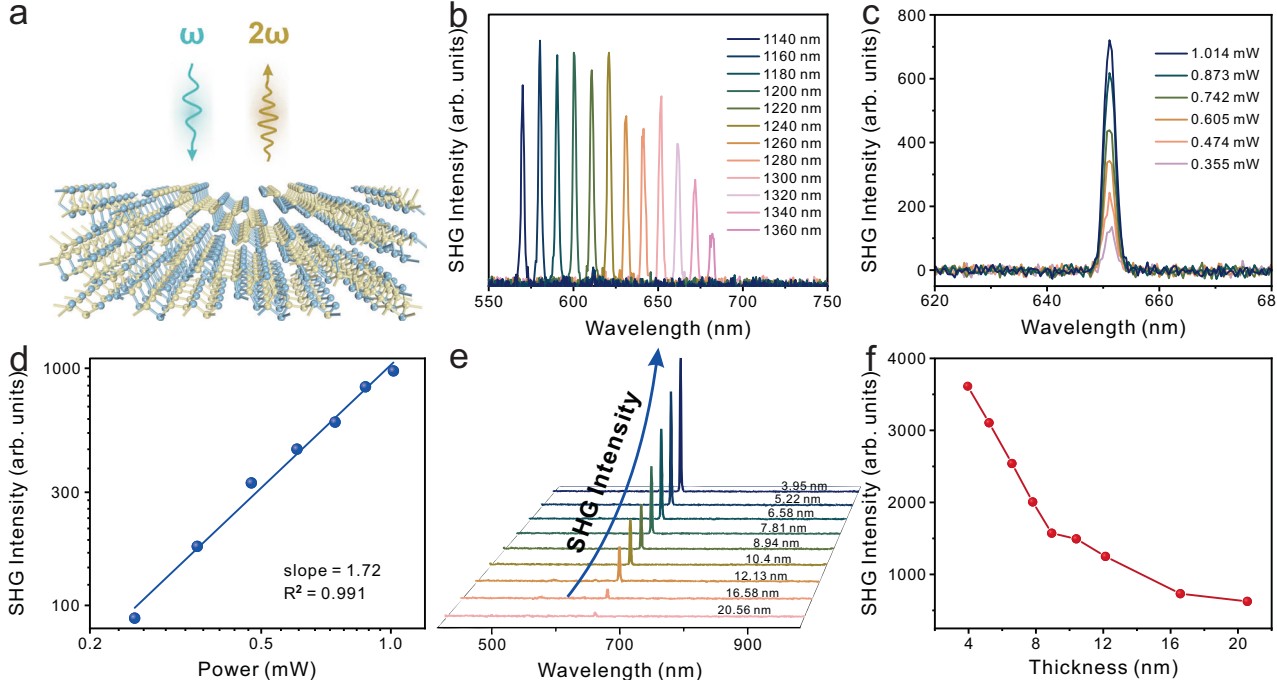

**Fig. 3 | Investigation of SHG in 2D QLDS-GaTe single crystal. a** Schematic representation of the SHG investigation conducted on the 2D QLDS-GaTe single crystal. The blue arrow refers to the incoming light with a frequency of $\omega$, and the yellow arrow refers to the second harmonic generated with a frequency of $2\omega$. Yellow atoms refer to Te atoms, and blue atoms refer to Ga atoms. **b** Wavelength-dependent investigation of SHG intensity under excitation power of 1 mW. **c** Dependency of SHG intensity on the excitation power. **d** Power-dependent SHG intensity plotted on logarithmic coordinates. **e** SHG spectra corresponding to each thickness of the sample. **f** Thickness-dependent SHG intensity relationship diagram extracted from (**e**).

on a 2D scale, distinguished by interlayer interactions that synergize vdW forces and covalent bonds. As a representative, the 2D QLDS-GaTe has been investigated, which manifests a skewed growth structure with a noticeable enhancement in interlayer coupling. This feature endows that lattice constant contraction along this orientation approaches the theoretical limit of 10.8%, bordering the 2D limit. It also exhibits remarkable anisotropy and pronounced SHG property, achieving a polarization tensor of 394.3 pm V$^{-1}$, hinting at potential applications in nonlinear optics.

Conclusively, the proposed concept of "2D quasi-layered materials with domino structures" effectively bridges the gap between traditional layered and non-layered materials. It ingeniously amalgamates the advantages of both material classes, facilitating the precise tuning of interlayer coupling effects. Moreover, its potential applicability extends across diverse fields, including optics, sensors, and catalysis. This approach provides a research platform for studying 2D materials, contributing to the field of material science.

## Methods
### CVD growth of 2D QLDS-GaTe
The 2D QLDS-GaTe single crystal was synthesized using chemical vapor deposition (CVD). Briefly, 7 mm × 7 mm tungsten (W) substrates were cleaned with piranha solution, isopropanol, and deionized water, sequentially. A 0.2 g Ga layer was applied on the cleaned substrate to form the growth substrate. This was placed at the center of a 10 cm × 2 cm quartz boat, which was situated mid-point in a quartz tube within the CVD furnace. Upstream, approximately 15 cm from the furnace center, Te powder was loaded into a separate quartz boat. The system was initially purged with Ar (500 sccm) for 2 min, followed by the continuous inflow of Ar (50 sccm) and H$_2$ (65 sccm). The W-Ga substrate alone was heated to 1080 °C for 40 min to initiate growth. Subsequently, the Te powder was introduced into the furnace, maintaining 1080 °C for 8 min. To preserve the initial growth stage, the

furnace was rapidly cooled to room temperature by opening the clam shell.

### Transfer of the sample
The transfer of the samples onto SiO$_2$/Si substrates was accomplished through the following steps: Initially, polymethyl methacrylate (PMMA) was spin-coated onto the W/G-GaTe samples at 600 rpm for 10 s, followed by 4000 rpm for 50 s, before baking on a hotplate at 180 °C for 4 min. Upon natural cooling, the sample was soaked in a 1:4 hydrochloric acid solution until Ga is completely reacted, detaching the GaTe flakes encapsulated in the PMMA film from the W substrate. Subsequently, the PMMA/GaTe layer was placed on the SiO$_2$/Si substrate. The PMMA was then removed by immersing the substrate in acetone at room temperature for 5 min, thus leaving the GaTe flakes on the SiO$_2$/Si substrate.

### The measurements of second harmonic generation
The power-dependent SHG measurements of 2D QLDS-GaTe were performed by a wavelength-tunable supercontinuum laser (NKT Photonics, EXB-4) with a frequency of 78 MHz. The second harmonic signal was collected by the spectral measurement path with a CCD camera (EXi Blue, Qimaging Inc.) and a spectrometer (iHR320, Horiba Jobin Yvon).

### Theoretical calculations
DFT calculations were conducted using the Vienna ab initio Simulation Package (VASP)[30] with the generalized gradient approximation (GGA) in the Perdew−Burke−Ernzerhof (PBE) form for the exchange-correlation functional. The projector-augmented wave (PAW) method was employed for the electron-ion interaction treatment[31]. A plane wave energy cutoff of 450 eV was set for the expansion of the Kohn−Sham orbitals. Convergence criteria for the force were defined as 0.02 eV Å$^{-1}$. To avoid spurious interactions

between periodic images, a vacuum layer of 20 Å was used. Grimme's D3 method with Becke–Jonson damping (DFT-D3) was applied for the dispersion correction to account for van der Waals interactions. The accuracy of these calculations was ensured by careful convergence testing and comparison with experimental data.

The Gibbs free energy of formation $\Delta G$ was evaluated as the function of the chemical potential of $\mu_{Te}$ shown below:

$$\Delta G = \left( G_{sys} - G_{sub} - N_1 \times E_{Ga} - N_2 \times \mu_{Te} \right) / A \qquad (1)$$

where $G_{sys}$ and $G_{sub}$ were the energy of the system and substrate, $E_{Ga}$ was the energy of the Ga atom in its bulk structure, $N_1/N_2$ was the number of Ga/Te atoms, $A$ was the area of the metal surface in the supercell calculation.

## Characterizations

OM images were performed by Olympus DX51. SEM images were obtained on a ZEISS Merlin Compact SEM. AFM images were measured on a confocal laser microscope system (Alpha 300RS+, WITec). The sample lamellas for the cross-sectional TEM characterization were prepared via FIB treatment by Helios 5 UC Dualbeam. TEM images were carried out on a JEM-ARM200F electron microscope, operating at 200 kV. Meanwhile, the elemental mapping of samples was collected by EDS. The STEM images were captured by a JEM-ARM200F electron microscope.

## Data availability

The data that support the findings of this study are available from the corresponding author upon request.

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

## Acknowledgements

The research was supported by the National Natural Science Foundation of China (Grant Nos. 22025303 and 21905210). We would like to acknowledge the Center for Electron Microscopy at Wuhan University for their substantial support in capturing images of single crystals. We thank the Core Facility of Wuhan University for the measurement of X-ray photoelectron spectroscopy. The numerical calculations in this paper have been done on the supercomputing system in the Supercomputing Center of Wuhan University.

## Author contributions

L.F. and M.Z. developed the concept and conceived the experiments. H.L., L.W., R.H., and S.H. carried out the main experiments. H.L., S.H., and L.W. wrote the paper. H.L. contributed to the theoretical calculations. J. Lin and S.Z. captured the SHG characterizations. J.Y. and J. Luo carried out the AFM characterization. Y.L. covered part of the HRTEM characterization. L.F. and M.Z. revised the paper. All the authors contributed to the data analysis and scientific discussion. J.G. and J.Z. cover part of STEM characterization.

## Competing interests

The authors declare no competing interests.
