## [Peer Review File · Nature Communications]

REVIEWER COMMENTS

Reviewer #1 (Remarks to the Author):

The work by H. Lan et al. reports a domino-structured 2D QLDG-GaTe acting as HER electrocatalyst. The authors claim that the quasi-layered domino-structure is unprecedented, which leads to remarkable anisotropy, second harmonic generation enhancement, as well as outstanding hydrogen evolution activity. Basically, the claimed structure is fancy and the exhibited activity indeed looks decent. However, the evidence used to support the formation of the 'domino structure' is not sufficient. Besides, the proposed 'Region Fold' concept is rather ambiguous. The elucidation of HER mechanism is also unlogic. Considering the high quality of nature communications, I cannot suggest publication before the following concerns having been properly addressed.

Concerns regarding the 'domino structure':

1. The current characterization data seems insufficient to support the author's claim of the proposed domino structure. It is necessary for the author to provide direct evidence of the atomic arrangement in both the planar and cross-sectional views to demonstrate this structure convincingly.
2. The optical microscopy images do not clearly indicate a two-dimensional structure, such as nanosheets or nanoribbons. From Figure S7, the material still appears crystalline rather than a thin two-dimensional structure. Optical microscopy images of samples with different thicknesses should be provided.
3. The scale of the AFM images does not correspond to the reported lateral dimensions, and data from AFM measurements of samples with different thicknesses should be included.
4. For ultra-thin samples, such as those with a thickness of 1.2 nm, high-magnification TEM images should be provided to reveal the fine morphological structure.
5. Assuming the sample indeed exhibits a domino structure as claimed by the author, it is important to investigate whether this synthesis method and structure can be extended to other material systems.

Concerns regarding the HER mechanism:

1. More explanations need to be provided to support the so-called 'Region Fold' theory. Basically, the statement that 'the reactive edge states project onto larger reactive surfaces, while inactive regions fold into non-reactive areas' is neither physically coherent nor chemically understandable. The authors claim that the DOS of domino GaTe is 'novel', but the explanation of its novelty is missing.
2. In Page 11 Line 227-228, it is necessary to explain why a surface 'acting as both a charge donor and acceptor' can 'facilitate catalytic reactions'. Specifically, it is of vital importance to elucidate the pertinence to HER.
3. The HER mechanism elucidated in Page 12 is not rational. In acidic conditions, HER proceeds via either Tafel or Heyrovsky mechanism, with adsorbed H acting as the only intermediate (Chem. Soc. Rev. 2014, 43, 6555). In this view, the stabilization of H* on unsaturated Ga site plays negative role in catalyzing HER.
4. Based on the free energy results in Figure 4d, Te should be the active sites in 2D QLDG-GaTe. I suggest the authors calculate the hydrogen adsorption free energy on edge-exposed Te sites in layered GaTe, and compare the calculated ΔG_{H^*} to justify the uniqueness of surface Te site in QLDG-GaTe against the edge Te site in layered GaTe.
5. It's not strange to see that poly GaTe and layered GaTe exhibit no activity for HER (Figure 4e), as neither Ga nor Te is active species for HER. Considering the activity enhancement of structure changed GaTe (domino-structured GaTe) is very huge, it is highly valuable to figure out a rational mechanism for this enhancement. I suggest the authors build two more domino-structured GaTe with different tilting angle to investigate the catalytic potential of this paradigm.
6. It's strange to see a negative charge distribution on adsorbed H atom since a Te-H σ -bond forms, as claimed by the authors. The authors should explain this. Besides, some statements seem to be raised by speculation. For example, the statement of 'the substrate can undergo minor lattice contraction to revert to its original stable bonding structure' is proposed with no evidence.
7. Throughout the section of HER performance, no reference is employed to support the discussions. The authors need to justify the discussions with solid evidence and support.

Reviewer #2 (Remarks to the Author):

NCOMMS-23-28213-T

This manuscript reports on the unique growth of GaTe in a quasi -2D layered fashion (domino orientation). The growth utilized liquid metal Ga as the substrate, which enables the growth mode resulting in thin films (where the thickness of the GaTe can be controlled).

The growth is characterized by numerous experimental techniques (HR-TEM, XRD, SIMS, etc...). Numerous computational simulations were performed to calculate crystal structure, band diagrams, charge density, etc... comparing the "domino" growth mode with bulk GaTe. The authors used optical techniques to measure a pronounced second harmonic generation enhancement, which is interesting for potential nonlinear optic applications.

Overall, this is a very thorough manuscript that presents an interesting growth mode for GaTe and presents potential applications. Figure 4 presents the utilization of this material for hydrogen evolution reactions. The HER experiments are interesting and almost seem like they should be in an independent study. I am not sure if the authors were compelled to include that data given the amount of data in the supplemental that is more aligned with a synthesis manuscript.

This data and manuscript should be published ... maybe separating out the HER as a separate study. I am not sure that this manuscript presents the level of novel impact that is expected for Nature Communications. GaTe is studied in many other systems. This study presents a unique and compelling new growth mode. Is this growth mode only unique for GaTe or can other systems be tuned to domino structures.

At the present I would recommend publication into a more specialized journal focused on material growth. However, I would not be opposed to the Editor or other referees coming to a different conclusion.

Reviewer #3 (Remarks to the Author):

This work has introduced a new concept, the 2D quasi-layered material, the interlayer of which is claimed to have synergistic blend of vdW forces and covalent bonds. The unique interlayer bonding and surface brings structural anisotropy, second harmonic generation enhancement, and hydrogen evolution reaction catalytic activity. The manuscript can be considered for publication after addressing the following issues.

1. The definition of a new material by the authors seems to be mainly due to the divergence of crystal growth from the planar substrate, which may not be accurate enough. I suggest the authors to give more information on the crystal structure itself, for example, layer registry, interlayer distance, and others, before defining a new material.
2. Besides of local electron microscopy characterization, it is also essential to provide macroscopic shape of the "single crystals".
3. The authors have published many papers with liquid metal as nuclei to grow different materials of varying shapes. What is the advancement or the critical point to achieve this skew growth?
4. Can the method reported in this work be used to grow other materials?

Reviewer #4 (Remarks to the Author):

The manuscript "2D Quasi-Layered Material with Domino Structure" by Lan and coauthors reports a new concept of 2D quasi-layered material, where the authors demonstrate GaTe as a representative material. The carefully tuned growth conditions enable the formation of 2D QLDS-type structure. Thanks to this unique structure, the authors found some performance improvements, such as second harmonic generation and hydrogen evolution reaction catalytic activity. The concept of structure model is new, and realization of outstanding performance would

be useful for several future applications. I am positive for the publication, but before that, I would like to ask the authors to address the following questions and comments.

-The structural data for 2D QLDS-GaTe is missing, such as lattice constants and fractional coordinates. As the authors imply, QLDS-GaTe is a quasi-layered structure, not the ideal layered structure. Then, what is the definition of "2D" in QLDS-GaTe? The conventional 2D model includes vacuum slab in the structure model, i.e., single carbon layer (graphene) or S-Mo-S tri-atomic layer (MoS₂). Does 2D QLDS-GaTe have vacuum slab?

-The authors calculated the phonon dispersive curves. I'm wondering whether 2D QLDS-GaTe is a new crystal polymorph of GaTe or it's same as already known structure, but just preferred crystal orientation is applied on the substrate. The vibration property of GaTe was reported, for example, Scientific Reports 11, 21202 (2021). Is this reported structure same as the model mentioned in this manuscript?

-The low-magnification TEM pictures in Fig.2 are hard to understand. The authors should indicate the schematic sample structure. Moreover, the width of the sample, where Ga and Te were detected, is as broad as 1 μm . This looks very thick in terms of 2D layer.

-Fig3e has no information on thickness. Please add legends.

-I'm curious to know the thickness dependence of FWHM of SHG spectra. Is it constant, or is there any dependence similar to SHG intensity?

-The improvement of the catalytic reactions is explained due to the presence of the edge states. This may be related to the non-bonded electrons at the surface atoms, which are absent in the conventional surface terminated 2D materials. If this is the case, the random oriented polycrystalline GaTe film could show a similar performance as it also possesses unsaturated edge states. Why was this enhancement observed only for the 2D QLDS-GaTe?

RESPONSE TO REVIEWERS' COMMENTS

To Referee #1

We wish to express our gratitude to the reviewer for the insightful comments and constructive suggestions provided on our manuscript. We have endeavoured to address your comments in a systematic and comprehensive manner, and the manuscript has been meticulously revised in line with your guidance. Please identify our revisions, which are distinctly marked in red.

It is our aspiration that this thoroughly revised version of the manuscript will facilitate better comprehension and offer enhanced informative value, thereby meeting the rigorous requirements for publication in *Nature Communications*.

Overall remark: *The work by H. Lan et al. reports a domino-structured 2D QLDS-GaTe acting as HER electrocatalyst. The authors claim that the quasi-layered domino-structure is unprecedented, which leads to remarkable anisotropy, second harmonic generation enhancement, as well as outstanding hydrogen evolution activity. Basically, **the claimed structure is fancy and the exhibited activity indeed looks decent**. However, the evidence used to support the formation of the 'domino structure' is not sufficient. Besides, the proposed 'Region Fold' concept is rather ambiguous. The elucidation of HER mechanism is also unlogic. Considering the high quality of nature communications, I cannot suggest publication before the following concerns having been properly addressed.*

Author reply: We deeply value your insightful feedback. In light of your suggestions, we have added more experimental data and theoretical simulations to further substantiate the quasi-layered material's domino structure. It's essential to clarify that the "Region Fold concept" pertains to a phenomenon observed in the past research of 2D materials' HER performance, not a theoretical assertion (*Adv. Mater.* **2016**, 28, 6197; *Nat. Commun.* **2019**, 10, 1217; *Nat. Commun.* **2022**, 13, 2193). Recognizing its significance, we've undertaken appropriate modifications to its description. Moreover, with your constructive guidance, we've reassessed and refined our approach to the HER mechanism, ensuring a more rigorous comprehension. We are committed to systematically addressing each of your concerns, aspiring to augment both the clarity and validity of our findings. Your suggestions undoubtedly elevate the academic standard of our research and we will systematically address each of your remarks as follows.

Comment 1: *The current characterization data seems insufficient to support the author's claim of the proposed domino structure. It is necessary for the author to provide direct evidence of the atomic*

arrangement in both the planar and cross-sectional views to demonstrate this structure convincingly.

Author reply: We deeply appreciate your valuable insights concerning material structural characterization. In accordance with your suggestion, we performed HAADF-STEM analysis on the material's cross-section. We confirmed the elemental composition, as shown in Figure R1 (also seen in Figure S15 in the revised manuscript). Moreover, the cross-sectional HAADF-STEM image of the sample explicitly display the material's pronounced domino structure, as depicted in Figure R2 (also seen in Figure S16 in the revised manuscript), thereby bolstering the rigor of our structural characterization.

Figure R1 | The EDS elemental mappings of the cross-section of the 2D QLDS-GaTe crystal.

Figure R2 | A HAADF-STEM image of the cross-section of the 2D QLDS-GaTe crystal.

Revised Manuscript: (Red text: Revisions in the manuscript)

These FFT patterns align closely with the calculated simulated FFT diffraction structures (Supplementary Fig. 14), corroborating the synthesis of GaTe with a domino structure on the (-101) plane. Furthermore, using cross-sectional HADDF-STEM, we directly observed a slanted atomic arrangement in the material, offering additional validation that the material is indeed 2D QLDS-GaTe with domino structure (Supplementary Fig. 15, 16).

Revised Supplementary Information: (Red text: Revisions in the Supplementary Information)

The EDS elemental mappings of the cross-section of the 2D QLDS-GaTe crystal. We deposited W as the protective layer and conducted cross-sectional EDS elemental mapping on the material. The results revealed a uniform elemental distribution of QLDS-GaTe crystal.

The HAADF-STEM image of the cross-section of the 2D QLDS-GaTe crystal. The cross-sectional HADDF-STEM images of the material depict a remarkably interface between 2D QLDS-GaTe and underlying Si substrate. Moreover, the cross-section of the crystal shows the domino structure and there's a notable correspondence with the atomic structure.

Furthermore, in the realm of material structure validation, atomic diffraction patterns serve as a compelling tool. In this manuscript, our primary focus gravitates toward the diffraction structure in the planar orientation of the given material. A notable alignment exists between the Fast Fourier Transform (FFT) outcomes derived experimentally and those extrapolated from atomistic theoretical models (Figure R3). Additionally, a harmonious correlation is evident between experimental and theoretical high-resolution transmission electron microscopy (HRTEM) findings (Figure R4). Such correlations robustly affirm the congruence of our synthesized material with its theoretical counterpart, suggesting that the material aligns closely with the projected domino structure.

Figure R3 | FFT patterns of 2D QLDS-GaTe under diverse thicknesses. **a**, Experimental and simulated FFT patterns of 2D QLDS-GaTe with the thickness of 1.2 nm. **b**, Experimental and simulated FFT patterns of 2D QLDS-GaTe with the thickness of about 10 nm.

Figure R4 | HRTEM images of 2D QLDS-GaTe under diverse thicknesses. **a**, Experimental and simulated HRTEM images of 2D QLDS-GaTe with the thickness of 1.2 nm. **b**, Experimental and simulated HRTEM images of 2D QLDS-GaTe with the thickness of about 10 nm.

Based on your suggestions, we have augmented Figure S14 by incorporating the atomistic structures used for simulation, thereby enhancing the clarity of the results, as depicted in Figure R5.

Figure R5 | Simulated FFT results of 2D QLDS-GaTe structure with different thicknesses.

Now, we have provided direct evidence of the atomic arrangement in both the planar and cross-sectional views to demonstrate this structure convincingly. We believe the newly added characterization data can better support our as-proposed domino structure.

Comment 2: *The optical microscopy images do not clearly indicate a two-dimensional structure, such as nanosheets or nanoribbons. From Figure S7, the material still appears crystalline rather than a thin two-dimensional structure. Optical microscopy images of samples with different thicknesses should be provided.*

Author reply: Thanks for your comment. In line with your suggestion, we have crafted optical microscopy (OM) images for samples with different thicknesses transferred to the SiO₂/Si substrate (Figure R6). We added Figure R6 to supplementary information as Figure S9.

Figure R6 | OM images of 2D QLDS-GaTe under diverse thicknesses.

Revised Manuscript: (Red text: Revisions in the manuscript)

Therefore, ISMG strategy with liquid metal can reduce the formation energy across an expansive chemical potential window tuned by temperatures (Supplementary Fig. 5–7) and facilitates an inclined crystal growth with a well-aligned growth orientation (Supplementary Fig. 8). Additionally, by extending the annealing duration, the concentration of Te dissolved in the liquid metal progressively increases, which in turn affects the thickness of the material (Supplementary Fig. 9, 10).

Revised Supplementary Information: (Red text: Revisions in the Supplementary Information)

OM images of 2D QLDS-GaTe under diverse thicknesses. As the annealing time increases, the amount of Te dissolved in the liquid metal also progressively rises, leading to an elevated chemical potential of Te. During the cooling process, the material tends to precipitate on the surface of the liquid metal. A higher concentration of Te accelerates this precipitation efficiency. While the material continues to exhibit a pronounced anisotropic lateral growth behavior, its vertical growth rate becomes increasingly significant. It's due to this characteristic that we can controllably synthesize 2D QLDS-GaTe of varying thicknesses. Therefore, by modulating the annealing duration, we can achieve the synthesis of a range of 2D QLDS-GaTe with diverse thicknesses.

Comment 3: *The scale of the AFM images does not correspond to the reported lateral dimensions, and data from AFM measurements of samples with different thicknesses should be included.*

Author reply: Thank you for your insightful guidance. The thickness of the synthesized 2D QLDS-GaTe can be modulated by varying the annealing duration, reaching a minimal thickness of 1.2 nm. It's noteworthy that the sample tested via AFM is distinct from the one examined using TEM, albeit with the same thickness. This leads to certain variations in the lateral dimensions. Moreover, heeding your suggestion, we have appended AFM results for samples of different thicknesses as outlined below, which has been added to supplementary information.

Figure R7 | AFM images of 2D QLDS-GaTe under diverse thicknesses.

Revised Supplementary Information: (Red text: Revisions in the Supplementary Information)

AFM images of 2D QLDS-GaTe under diverse thicknesses. AFM analysis further corroborates that by modulating the annealing duration, we can synthesize 2D QLDS-GaTe samples with a diverse range of thicknesses.

Comment 4: *For ultra-thin samples, such as those with a thickness of 1.2 nm, high-magnification TEM images should be provided to reveal the fine morphological structure.*

Author reply: We sincerely value your constructive suggestions. The “Thinner Sample” referenced in our TEM analysis was derived using growth conditions established for a 1.2 nm sample. Both its low-magnification and high-magnification TEM images are depicted in Figure R8, correlating with the results in Figure 2 of the main text. The HRTEM micrographs obtained from our experiments align remarkably well with the predicted structures, notably manifesting a distinct lattice contraction. Such an alignment not only resonates with the configurations depicted in Figure R3 but is also congruent with the simulated results of structures approximately 1 nm in thickness as presented in the same figure. In light of these observations, we have bolstered our manuscript to uphold a greater standard of scientific precision.

Figure R8 | Low-magnification, high-magnification TEM, and simulated high-magnification TEM images of 2D QLDS-GaTe samples with the thickness of 1.2 nm.

Revised Manuscript: (Red text: Revisions in the manuscript)

Notably, the interlayer (101) spacing in the high-resolution transmission electron microscopy (HRTEM) images of samples with various thicknesses were delineated (Fig. 2d, 2g), measuring 7.33 Å for the thin sample (~1 nm) and 8.06 Å for the thicker one (~10 nm).

Revised Supplementary Information: (Red text: Revisions in the Supplementary Information)

Simulated FFT results of 2D QLDS-GaTe structure with different thicknesses. As the material thickness diminishes and interlayer coupling effects intensify, the diffraction patterns of 2D QLDS-GaTe correspondingly transform. We simulated the FFT patterns for structures approximately 1 nm and 4 nm thick, derived from our calculations, as shown in the figure below. **Given that the lattice constant of the material doesn't exhibit significant variations beyond 2 nm, and the simulated crystal diffraction patterns and interplanar spacings also show no pronounced differences at thicknesses beyond this value, we chose a 4 nm thick model to simulate the diffraction structure characteristic of a bulk crystal.** As the interlayer coupling effect intensifies, a significant contraction in the interplanar spacing of the (101) crystal plane becomes apparent, indicating the enhancement of interlayer coupling effects.

Comment 5: *Assuming the sample indeed exhibits a domino structure as claimed by the author, it is important to investigate whether this synthesis method and structure can be extended to other material systems.*

Author reply: Your insightful feedback has been instrumental. Guided by your remarks, we delved deeper into the robustness of our growth procedure, revealing a new quasi-layered material with a domino configuration.

Figure R9 | . **a, b**, Side view and top view of 2D QLDs-In₄Te₃ structure. **c**, A low magnification TEM image of 2D QLDs-In₄Te₃ sample and EDS elemental mappings of Te and In. **d, e**, FFT patterns and HRTEM image of 2D QLDs-In₄Te₃ sample.

Figure R9a–b illustrates this unique structure, showcasing an inclined arrangement in atomic orientation. Notably, the material's interlayer forces occupy a niche between vdW interactions and covalent bonds, directly echoing our conception of QLDs materials. Turning to the chemical characterization of the sample, we deployed EDS analysis. As detailed in Figure R9c, In₄Te₃ flakes are discerned as the primary component. Further substantiating this, EDS elemental mapping consistently portrays an even In/Te distribution across the In₄Te₃ flake, confirming material uniformity. The material's crystalline nature is irrefutably evidenced by the singular diffraction patterns in FFT images (Figure R9d) and further corroborated by the HRTEM depiction in Figure R9e. Here, distinct interplanar spacings, denoted as (200) and (002), elucidate that the nascent In₄Te₃ samples favour growth along the (040) axis, in alignment with structural elucidation in Figure R9a–b. Such discoveries not only attest to the versatility of our synthesis technique for myriad domino-structured quasi-layered materials but also refine and fortify the foundation for subsequent investigations in this field.

Revised Manuscript: (Red text: Revisions in the manuscript)

Therefore, ISMG strategy with liquid metal can reduce the formation energy across an expansive chemical potential window tuned by temperatures (Supplementary Fig. 5–7) and facilitates an inclined crystal growth with a well-aligned growth orientation (Supplementary Fig. 8). Additionally, by extending the annealing duration, the concentration of Te dissolved in the liquid metal progressively increases, which in turn affects the thickness of the material (Supplementary Fig. 9, 10). **The ISMG**

strategy exhibits universality. Through this approach, we've also synthesized indium telluride materials with a domino structure (Supplementary Fig. 11). This discovery not only broadens the applicability of the method but also lays a foundation for unveiling more 2D QLDS materials.

Revised Supplementary Information: (Red text: Revisions in the Supplementary Information)

Structural characterizations of 2D QLDS-In₄Te₃ samples. Supplementary Figure 11a,b illuminate the distinct architecture of the material, capturing a notably tilted atomic configuration. The interlayer interactions characterizing this material intriguingly bridge the gap between vdW forces and covalent bonds, precisely resonating with the quintessential traits of QLDS materials. Transitioning to its chemical profile, we harnessed EDS techniques. Evident in Supplementary Figure 11c, the EDS elemental mapping showcases a homogenous In/Te distribution within the In₄Te₃ domain, underscoring its compositional consistency. The crystalline integrity of the material is unambiguously manifested in the singular diffraction patterns encapsulated in the FFT visuals in Supplementary Figure 11d, and this assertion gains further traction through the high-resolution TEM image depicted in Supplementary Figure 11e. Herein, prominent lattice spacings, tagged as (200) and (002), reveal a predilection of nascent In₄Te₃ samples for growth along the (040) facet, harmonizing with structural narratives in Supplementary Figure 11a–b. These revelations not only vouch for the adaptability of our fabrication approach to a gamut of domino-structured quasi-layered materials but also bolster the edifice for future explorations in this intriguing arena.

Comment 6: *More explanations need to be provided to support the so-called 'Region Fold' theory. Basically, the statement that 'the reactive edge states project onto larger reactive surfaces, while inactive regions fold into non-reactive areas' is neither physically coherent nor chemically understandable. The authors claim that the DOS of domino GaTe is 'novel', but the explanation of its novelty is missing.*

Author reply: We are profoundly grateful for your constructive feedback. On reflection, we concede that our initial designation of “theory” was misapplied; “concept” is decidedly more apt. Within the sphere of HER studies involving 2D layered materials, a compelling narrative has emerged: the edges of such materials consistently outpace the basal planes in terms of catalytic efficiency (*Nat. Mater.* **2016**, *15*, 1003; *Adv. Mater.* **2022**, *34*, 2202479; *Adv. Mater.* **2017**, *29*, 1701955). This edge-centric superiority served as the catalyst for our development of original “Region Fold” concept. Notably, the architecture of 2D QLDS materials beautifully dovetails with our conceptual framework—by maximizing the exposure of these highly reactive edges, these materials unambiguously elucidate the

underpinnings of their catalytic prowess.

Based on this phenomenon, we rename Region Fold theory as Region Fold concept. Meanwhile, the statement of “the reactive edge states project onto larger reactive surfaces, while inactive regions fold into non-reactive areas” has been deleted, thus ensuring the accuracy.

Furthermore, in our investigation of the 2D QLDS-GaTe electronic structure, we identified an intriguing deviation in its DOS from that of traditional layered GaTe, a distinction underscored in Figure R10. This deviation can be attributed to the 2D QLDS-GaTe’s unique growth orientation, resulting in surface characteristics distinct from its layered counterpart. Of particular note is a prominent peak at the Fermi energy in the DOS of 2D QLDS-GaTe, a feature conspicuously absent in layered GaTe. Such an electronic signature suggests that 2D QLDS-GaTe might not conform to typical semiconductor behaviors, potentially leaning toward metallic properties. It is this distinctiveness that prompts us to label this DOS pattern as “novel”.

Figure R10 | Computed DOS for the 2D layered GaTe and 2D QLDS-GaTe configuration.

In alignment with your insights, we have meticulously revised our content, which is presented below.

Revised Manuscript: (Red text: Revisions in the manuscript)

HER performance of 2D QLDS-GaTe single crystal. Previous research underscores the superior catalytic performance of unsaturated edges in layered materials, evidenced by 2D MoS₂ catalysis (Supplementary Fig. 25). **In layered materials, catalytic activity markedly varies across different regions. Specifically, the basal plane tends to be catalytically lackluster compared to the more reactive edges (Supplementary Fig. 26, 27, 35, 36). (Nat. Mater. 2016, 15, 1003; Adv. Mater. 2022, 34, 2202479; Adv. Mater. 2017, 29, 1701955)** These edge states favor catalytic reactions due to their unique charge characteristics. **From this observation emerges a salient hypothesis: what if reconfiguring the material,**

amplifying the exposure of the reactive edge while relegating the less active basal plane to regions less critical to the reaction? Guided by this principle, we delved into the architecture of 2D QLDS-GaTe. This structure adeptly brings the unsaturated reactive edges of layered GaTe to the surface of 2D QLDS-GaTe while sequestering the catalytically passive saturated basal plane of layered GaTe deeper into the bulk of 2D QLDS-GaTe. In short, for domino-structured materials, a skewed growth pattern projects the unsaturated edges of the layered structure onto the surface of the domino-structured material, thereby not only augmenting the so-called edge area but also endowing the material with a novel density of states (DOS), as depicted in Fig. 4b, which suggests that 2D QLDS-GaTe might not conform to typical semiconductor behaviors, potentially leaning towards metallic properties. This transformation is coined as the “Region Fold” concept. Guided by this concept, more pronounced surface charge polarization occurs, with the surface acting as both a charge donor and acceptor (Fig. 4c, Supplementary Fig. 28–30), facilitating catalytic reactions.

Comment 7: *In Page 11 Line 227-228, it is necessary to explain why a surface ‘acting as both a charge donor and acceptor’ can ‘facilitate catalytic reactions’. Specifically, it is of vital importance to elucidate the pertinence to HER.*

Author reply: We greatly appreciate your insightful guidance and recommendations. While electron acceptors are pivotal in driving the transformation of H^+ to H^* , electron donors chiefly mediate the progression from H^* to H_2 . Notably, within the realm of 2D QLDS-GaTe, the Te sites take center stage as dominant electron donors. Their distinctive localized electronic states serve as a catalyst, profoundly expediting the HER mechanism as highlighted in a recent study (*Nat. Commun.* **2022**, *13*, 2193).

Utilizing a partial charge density plot allows us to intricately map the charge state distributions across distinct energy regimes, as elegantly showcased in Figure R11.

Figure R11 | Partial charge density plots across varying energy ranges.

The occupied orbitals below the Fermi energy level, displayed in the left figure, reveal a discernible charge density distribution across both Ga and Te atoms on the surface of 2D QLDS-GaTe. This distribution suggests that these electrons are poised to be donated to adsorbates, classifying the surface as a potential Electron Donor. On the other hand, the right figure delineates the empty orbitals situated above the Fermi energy. Here, both Ga and select Te atoms feature orbitals at the Fermi level, ready to accept electrons from adsorbed species. This positions them as Electron Acceptors. Taken together, these observations illuminate the unique duality of the 2D QLDS-GaTe surface, capable of functioning simultaneously as both Electron Donor and Electron Acceptor.

Revised Supplementary Information: (Red text: Revisions in the Supplementary Information)

Partial charge density plots across varying energy ranges. The partial charge density plot results suggest that Te atoms on the surface of 2D QLDS-GaTe can function as atomic donors, facilitating the progression of the catalytic reaction. Surface Ga atoms can serve as both atomic donors and acceptors, thus stabilizing the adsorbed H*. The synergistic effect of these two types of atoms is fundamental to the ultra-low overpotential observed in 2D QLDS-GaTe. **The occupied orbitals below the Fermi energy level, displayed in the left figure, reveal a discernible charge density distribution across both Ga and Te atoms on the surface of 2D QLDS-GaTe. This distribution suggests that these electrons are poised to be donated to adsorbates, classifying the surface as a potential Electron Donor. On the other hand, the right figure delineates the empty orbitals situated above the Fermi energy. Here, both Ga and select Te atoms feature orbitals at the Fermi level, ready to accept electrons from adsorbed species. This positions them as Electron Acceptors. Taken together, these observations illuminate the unique duality of the 2D QLDS-GaTe surface, capable of functioning simultaneously as both Electron Donor and Electron Acceptor.**

Comment 8: *The HER mechanism elucidated in Page 12 is not rational. In acidic conditions, HER proceeds via either Tafel or Heyrovsky mechanism, with adsorbed H acting as the only intermediate (Chem. Soc. Rev. 2014, 43, 6555). In this view, the stabilization of H* on unsaturated Ga site plays negative role in catalyzing HER.*

Author reply: We are grateful for your astute observations. Our rationale behind detailing the Ga site was to engender a holistic exploration, prompting us to delve into both the surface-exposed Ga and Te. We contend that Ga does not exhibit pronounced reactivity. Once the Ga sites on the surface achieve adsorption saturation, the activity of the Te sites progressively takes precedence, thereby facilitating the progression of the HER reaction. Yet, in championing a thorough investigation, we contend that

elucidating the reactivity of both Ga and Te sites will more lucidly spotlight the pivotal role of surface Te atoms in HER reactivity, overshadowing that of Ga. To preclude potential misconceptions surrounding the Ga site's reactivity, we have instituted the subsequent amendments:

Revised Manuscript: (Red text: Revisions in the manuscript)

Gibbs free energy profiles in Fig. 4d affirm this theory. The highly active Te sites effectively promote the conversion of H^+ to H_2 , recognized as a critical factor in HER performance, whereas the unsaturated Ga atoms on the surface of the 2D QLDS-GaTe structure exhibit less reactivity (*Chem. Soc. Rev.* **2014**, *43*, 6555). (Fig. 4d, Supplementary Fig. 31–34). Moreover, the reactivity of Te atoms surpasses that of layered GaTe (Supplementary Fig. 35, 36), with ΔG^* concentrated in the range of -0.5 eV to 0.5 eV, which is highly conducive to the HER process. This further validates the applicability of the Region Fold theory in domino-structure materials.

Comment 9: *Based on the free energy results in Figure 4d, Te should be the active sites in 2D QLDS-GaTe. I suggest the authors calculate the hydrogen adsorption free energy on edge-exposed Te sites in layered GaTe, and compare the calculated ΔG_{H^*} to justify the uniqueness of surface Te site in QLDS-GaTe against the edge Te site in layered GaTe.*

Author reply: We deeply value your constructive feedback. In light of your suggestions, we've expanded our analysis to elucidate the HER activity at the edges of layered GaTe, as depicted in Figure R12 and R13. It is noteworthy that, for layered GaTe, the Te sites within the edge region demonstrate a pronouncedly superior HER catalytic activity when juxtaposed with those in the basal plane, which are encumbered by a reaction barrier of 1.75 eV. The inclusion of this additional data, as prompted by your feedback, not only reinforces the credibility of our study but also accentuates the pivotal role of the 2D QLDS materials' design concept. By adopting this design principle, we capitalize on maximizing the exposure of the more reactive edge regions, which are intrinsically present in layered structures. This strategic exposure unequivocally drives the enhancement in the overall catalytic performance of the material.

Figure R12 | Calculated Gibbs free energy profiles of the edge region of 2D layered GaTe.

Figure R13 | Structures of potential HER activation sites on 2D layered GaTe edge.

Revised Manuscript: (Red text: Revisions in the manuscript)

HER performance of 2D QLDS-GaTe single crystal. Previous research underscores the superior catalytic performance of unsaturated edges in layered materials, evidenced by 2D MoS₂ catalysis (Supplementary Fig. 25). In layered materials, catalytic activity markedly varies across different regions. Specifically, the basal plane tends to be catalytically lackluster compared to the more reactive edges. (*Nat. Mater.* **2016**, *15*, 1003; *Adv. Mater.* **2022**, *34*, 2202479; *Adv. Mater.* **2017**, *29*, 1701955) **This trend is also evident in 2D layered GaTe, where the ΔG^* at the edge region markedly outperforms that within the plane (Supplementary Fig. 26, 27, 35, 36).**

Revised Supplementary Information: (Red text: Revisions in the Supplementary Information)

Calculated Gibbs free energy profiles of the edge region of 2D layered GaTe. Computational

findings indicate that the Gibbs free energy of the edge region is significantly more favorable than that of the material's basal plane.

Structures of potential HER activation sites on 2D layered GaTe edge. Computational results suggest that Te sites at the edge region exhibit robust catalytic activity, facilitating the progression of the HER reaction.

Comment 10: *It's not strange to see that poly GaTe and layered GaTe exhibit no activity for HER (Figure 4e), as neither Ga nor Te is active species for HER. Considering the activity enhancement of structure changed GaTe (domino-structured GaTe) is very huge, it is highly valuable to figure out a rational mechanism for this enhancement. I suggest the authors build two more domino-structured GaTe with different tilting angle to investigate the catalytic potential of this paradigm.*

Author reply: We are genuinely appreciative of your invaluable input. We postulate that the distinct electronic states at the Fermi level in 2D QLDS-GaTe, when compared to its layered GaTe counterpart, might play a pivotal role in enhancing the catalytic reactions (Figure R10). To ensure that this enhancement is not a peculiarity of a specific orientation, we've modelled various tilt angles in line with your recommendations (61.9° , 38.5°), and the outcomes are presented in Figure R14–17. It's evident that while there are variations in catalytic performance with changing angles, all configurations of 2D QLDS-GaTe outperform the layered GaTe. The incorporation of these computational insights broadens our research vista, laying a robust foundation for future exploration into the catalytic activities of 2D QLDS-GaTe structures at different orientations.

Figure R14 | Calculated Gibbs free energy profiles of 2D QLDS-GaTe under the tilt angle of 61.9° .

Figure R15 | Structures of potential HER activation sites on 2D QLDS-GaTe under the tilt angle of 61.9°.

Figure R16 | Calculated Gibbs free energy profiles of 2D QLDS-GaTe under the tilt angle of 38.5°.

Figure R17 | Structures of potential HER activation sites on 2D QLDS-GaTe under the tilt angle of 38.5°.

Revised Manuscript: (Red text: Revisions in the manuscript)

Moreover, the reactivity of Te atoms surpasses that of layered GaTe (Supplementary Fig. 35, 36), with ΔG^* concentrated in the range of -0.5 eV to 0.5 eV, which is highly conducive to the HER process. This further validates the applicability of the Region Fold concept in domino-structure materials. Furthermore, we investigated the HER performance of Te in 2D QLDS-GaTe structures with varying tilt angles. As depicted in Supplementary Fig. 37–40, even with changes in the inclination angle, the ΔG^* corresponding to the Te atoms consistently outperforms that of layered GaTe. This discovery paves the way for future research into 2D QLDS-GaTe with different tilt angles.

Revised Supplementary Information: (Red text: Revisions in the Supplementary Information)

Calculated Gibbs free energy profiles of 2D QLDS-GaTe under the tilt angle of 61.9° . We constructed a model with a tilt angle of 61.9° and subsequently computed the HER performance of surface Te atoms. The results indicate that the HER activity of Te atoms at this inclination significantly surpasses that of layered GaTe.

Structures of potential HER activation sites on 2D QLDS-GaTe under the tilt angle of 61.9° . The structures below correspond to the active sites involved in the reaction, as shown in Supplementary Figure 37.

Calculated Gibbs free energy profiles of 2D QLDS-GaTe under the tilt angle of 38.5° . We constructed a model with a tilt angle of 38.5° and subsequently calculated the HER performance of surface Te atoms. The results reveal that the HER activity of Te atoms under this tilt angle markedly surpasses that of layered GaTe.

Structures of potential HER activation sites on 2D QLDS-GaTe under the tilt angle of 38.5° . The structures below correspond to the active sites involved in the reaction, as shown in Supplementary Figure 39.

Comment 11: *It's strange to see a negative charge distribution on adsorbed H atom since a Te-H σ -bond forms, as claimed by the authors. The authors should explain this. Besides, some statements seem to be raised by speculation. For example, the statement of 'the substrate can undergo minor lattice contraction to revert to its original stable bonding structure' is proposed with no evidence.*

Author reply: We appreciate the insightful feedback provided. For layered GaTe, a Te–H σ -bond formation is evident, yet it exhibits relative weakness, a detail discernible from an overhead perspective as shown in Figure R18. Our computational analysis highlights a modest charge accumulation at the Te–H bond. Interestingly, when juxtaposed with the findings from 2D QLDS-GaTe, the accumulation is

less pronounced, an observation that aligns with our COHP computational results. Moreover, a pronounced negative charge distribution is evident in the layered GaTe computations. This phenomenon can be attributed to the partial occupancy of the anti-bonding orbitals in the Te–H linkage, as detailed in Table R1.

Besides, based on your suggestion, we've also made revisions to certain descriptions that were deemed inappropriate, enhancing both the rigor and scientific novelty of our study.

Figure R18 | Charge density difference maps with H* adsorption. a, b, Top view of charge density difference plots of 2D QLDS-GaTe and layered GaTe with H* adsorption corresponding to Fig. 4g, 4h.

Table R1. Calculated bonding state of the H adsorption on 2D layered GaTe structure corresponding to Fig. 4h.

NBO	Occupancy	Center (bond contribution, %)		Hybridization (Function, %)	
		Te	H	Te	H
Te–H (σ)	1.937	Te (54.47)	H (45.53)	Te ($s^{1.77}p^{97.96}d^{0.27}$)	H (s^{100})
Te–H (σ^*)	0.275	Te (45.53)	H (54.47)	Te ($s^{1.77}p^{97.96}d^{0.27}$)	H (s^{100})

Revised Manuscript: (Red text: Revisions in the manuscript)

The crystal orbital Hamilton population (COHP) results underscore the superior capacity of 2D QLDS-GaTe for H* capture, while the layered GaTe shows a relatively weaker interaction with a partially filled σ^* antibonding orbital. This discrepancy, further corroborated by the natural population analysis (NPA) charge of H, accounts for the high ΔG^* inherent to layered GaTe. In 2D QLDS-GaTe, Ga atoms linked to Te compete with H for electron donation (Supplementary Fig. 44), resulting in the weakening of the original Te–Ga bonds. Under these circumstances, the substrate **tends to revert to its original stable bonding structure**, which leads to the shortening of the Te–Ga distance and consequently increases (reduces) the electron contribution from Ga (Te) (*ACS Nano* **2019**, *13*, 11874). This sequence of events results in a smaller ΔG^* during the HER reaction, which is a key factor underlying the superior HER performance of 2D QLDS-GaTe.

Comment 12: *Throughout the section of HER performance, no reference is employed to support the discussions. The authors need to justify the discussions with solid evidence and support.*

Author reply: Thank you for your insightful suggestions. We have incorporated several highly relevant references into the manuscript to enhance the clarity and scientific rigor of our results and discussions.

Revised Manuscript: (Red text: Revisions in the manuscript)

1) In layered materials, catalytic activity markedly varies across different regions. Specifically, the basal plane tends to be catalytically lackluster compared to the more reactive edges. (*Nat. Mater.* **2016**, *15*, 1003; *Adv. Mater.* **2022**, *34*, 2202479; *Adv. Mater.* **2017**, *29*, 1701955)

2) These edge states favor catalytic reactions due to their unique charge characteristics. (*Nat. Nanotechnol.* **2016**, *11*, 218; *Adv. Energy Mater.* **2022**, *12*, 2003841)

3) In short, for domino-structured materials, a skewed growth pattern projects the unsaturated edges of the layered structure onto the surface of the domino-structured material, thereby not only augmenting the so-called edge area but also endowing the material with a novel density of states (DOS), as depicted in Fig. 4b, which suggests that 2D QLDS-GaTe might not conform to typical semiconductor behaviors, potentially leaning towards metallic properties. (*Adv. Sci.* **2020**, *7*, 2002172)

4) The unsaturated Ga atoms on the surface of the 2D QLDS-GaTe structure favor the stabilization of H*, impeding optimal HER catalysis (*Chem. Soc. Rev.* **2014**, *43*, 6555), while the highly active Te sites promote the conversion of H* to H₂ (Fig. 4d, Supplementary Fig. 31–34).

5) Under these circumstances, the substrate tends to revert to its original stable bonding structure, which leads to the shortening of the Te–Ga distance and consequently increases (reduces) the electron contribution from Ga (Te). (*ACS Nano* **2019**, *13*, 11874)

To Referee #2

We are grateful for the thoughtful comments and suggestions provided by the reviewer. Each comment has been addressed comprehensively, and we have undertaken revisions in accordance with your recommendations. Our amendments can be located in the manuscript, highlighted in red. It is our aspiration that these modifications have enhanced the clarity and depth of the manuscript, aligning it with the publication standards of *Nature Communications*.

Overall remark: *This manuscript reports on the unique growth of GaTe in a quasi-2D layered fashion (domino orientation). The growth utilized liquid metal Ga as the substrate, which enables the growth mode resulting in thin films (where the thickness of the GaTe can be controlled).*

The growth is characterized by numerous experimental techniques (HR-TEM, XRD, SIMS, etc...). Numerous computational simulations were performed to calculate crystal structure, band diagrams, charge density, etc... comparing the "domino" growth mode with bulk GaTe. The authors used optical techniques to measure a pronounced second harmonic generation enhancement, which is interesting for potential nonlinear optic applications.

Overall, this is a very thorough manuscript that presents an interesting growth mode for GaTe and presents potential applications. Figure 4 presents the utilization of this material for hydrogen evolution reactions. The HER experiments are interesting and almost seem like they should be in an independent study. I am not sure if the authors were compelled to include that data given the amount of data in the supplemental that is more aligned with a synthesis manuscript.

Author Reply: We extend our profound gratitude for your acknowledgment of our research and the valuable insights you've provided. Rooted in the existing framework of two-dimensional (2D) materials, our pursuit is aimed at identifying a novel family of materials that bridges the divide between traditional layered and non-layered counterparts. Ever since the emergence of 2D material systems in 2004, they have undergone a remarkable evolution. Historically, these 2D materials have predominantly unveiled themselves in two archetypal forms: layered entities (such as graphene, h-BN, TMDs, and the like) and their non-layered counterparts (including GaN, CeO₂, Mo₂GaC, among others). This bifurcation largely arises from the potency of the interlayer coupling effect. Intrinsically, the atomic disposition within these materials is typified by a densely packed orientation, strategically aligning to mitigate entropy and amplify structural tenacity. Such a layout inherently fosters an orthogonal interplay between the interlayer interactions and the 2D growth plane, consequently negating any latent predisposition for material deformation or fragmentation.

Figure R1 | The schematic diagram of layered quasi-layered, and non-layered material.

Building upon this foundational understanding, we unveil an uncharted material system christened as the 2D quasi-layered material with a domino structure. This stands distinctly apart from both layered and non-layered materials, offering a fresh perspective within the 2D material paradigm (Figure R1). Evocative of the close-knit arrangement of domino tiles, these 2D quasi-layered materials are characterized by a slanted growth mode. To capture its essence, we have carefully termed it as the 2D quasi-layered domino-structured (QLDS) materials, shedding light on its singular position in the vast tapestry of 2D materials.

The distinctiveness of this structure invariably raises a question in many researcher's mind: does it bear superior catalytic properties? Drawing from previous studies on 2D layered materials, the catalytic activity of edge regions has consistently surpassed that of in-plane areas (*Nat. Mater.* **2016**, *15*, 1003; *Adv. Mater.* **2022**, *34*, 2202479; *Adv. Mater.* **2017**, *29*, 1701955). The emergence of 2D QLDS-GaTe appears to maximize the extension of these catalytically potent edge regions, thus hinting at its potential to exhibit exceptional catalytic performance. Moreover, while intercalation modifications of layered materials often demand stringent external conditions (*Nat. Commun.* **2021**, *12*, 5886), the advent of 2D QLDS materials undeniably offers a groundbreaking platform to address this challenge.

Given the comprehensiveness of the study and the heightened interest in this sector, we firmly believe that the electrocatalysis aspect of our research is indispensable. Furthermore, it warrants a more in-depth exploration to pave the way for future investigators in this realm. Beyond that, we are committed to expanding the universality of material synthesis, thereby broadening the 2D QLDS materials portfolio. This initiative aims to lay a solid foundation for subsequent investigations into this burgeoning material family.

Comment 1: *This data and manuscript should be published ... maybe separating out the HER as a separate study. I am not sure that this manuscript presents the level of novel impact that is expected for Nature Communications. GaTe is studied in many other systems. This study presents a unique and compelling new growth mode. Is this growth mode only unique for GaTe or can other systems be tuned to domino structures?*

Author Reply: We greatly appreciate your insightful suggestions. Our research focuses on a class of materials that bridges the divide between layered and non-layered materials, termed “quasi-layered materials”. The discovery of these materials effectively fills the gap between layered and non-layered entities, offering a fresh perspective for structural modification studies in 2D materials. The distinctiveness of this structure invariably raises a question in many researcher’s mind: does it bear superior catalytic properties? The emergence of 2D QLDS-GaTe appears to maximize the extension of these catalytically potent edge regions, thus hinting at its potential to exhibit exceptional catalytic performance. Moreover, while intercalation modifications of layered materials often demand stringent external conditions, the advent of 2D QLDS materials undeniably offers a groundbreaking platform to address this challenge. Based on this, we have explored the HER performance to further elucidate the enormous potential of the 2D QLDS materials we have discovered in future applications.

Furthermore, it's worth noting that our synthesis approach is not exclusive to GaTe but has broader applicability. Guided by your remarks, we delved deeper into the robustness of our growth procedure, revealing a new quasi-layered material with a domino configuration. We have successfully extended our method to the synthesis of 2D In_4Te_3 with the domino structure.

Figure R2 | Structural characterizations of 2D QLDS- In_4Te_3 samples. a, b, Side view and top view of 2D QLDS- In_4Te_3 structure. **c,** A low magnification TEM image of 2D QLDS- In_4Te_3 sample and EDS elemental mappings of Te and In. **d, e,** FFT patterns and HRTEM image of 2D QLDS- In_4Te_3 sample.

Figure R2a–b illustrates this unique structure, showcasing an inclined arrangement in atomic orientation. Notably, the material’s interlayer forces occupy a niche between vdW interactions and covalent bonds, directly echoing our conception of QLDS materials. Turning to the chemical characterization of the sample, we deployed EDS analysis. As detailed in Figure R2c, In_4Te_3 flakes are discerned as the primary component. Further substantiating this, EDS elemental mapping consistently portrays an even In/Te distribution across the In_4Te_3 flake, confirming material uniformity. The material’s crystalline nature is irrefutably evidenced by the singular diffraction patterns in FFT images (Figure R2d) and further corroborated by the HRTEM depiction in Figure R2e. Here, distinct interplanar spacings, denoted as (200) and (002), elucidate that the nascent In_4Te_3 samples favour growth along the (040) axis, in alignment with structural elucidation in Figure R2a–b. Such discoveries not only attest to the versatility of our synthesis technique for myriad domino-structured quasi-layered materials but also refine and fortify the foundation for subsequent investigations in this field.

Revised Manuscript: (Red text: Revisions in the manuscript)

Therefore, ISMG strategy with liquid metal can reduce the formation energy across an expansive chemical potential window tuned by temperatures (Supplementary Fig. 5–7) and facilitates an inclined crystal growth with a well-aligned growth orientation (Supplementary Fig. 8). Additionally, by extending the annealing duration, the concentration of Te dissolved in the liquid metal progressively increases, which in turn affects the thickness of the material (Supplementary Fig. 9, 10). **The ISMG strategy exhibits universality. Through this approach, we’ve also synthesized indium telluride materials with a domino structure (Supplementary Fig. 11). This discovery not only broadens the applicability of the method but also lays a foundation for unveiling more 2D QLDS materials.**

Revised Supplementary Information: (Red text: Revisions in the Supplementary Information)

Structural characterizations of 2D QLDS- In_4Te_3 samples. Supplementary Figure 11a,b illuminate the distinct architecture of the material, capturing a notably tilted atomic configuration. The interlayer interactions characterizing this material intriguingly bridge the gap between vdW forces and covalent bonds, precisely resonating with the quintessential traits of QLDS materials. Transitioning to its chemical profile, we harnessed EDS techniques. Evident in Supplementary Figure 11c, the EDS elemental mapping showcases a homogenous In/Te distribution within the In_4Te_3 domain, underscoring its compositional consistency. The crystalline integrity of the material is unambiguously manifested in the singular diffraction patterns encapsulated in the FFT visuals in Supplementary Figure 11d, and this assertion gains further traction through the high-resolution TEM image depicted in Supplementary Figure 11e. Herein, prominent lattice spacings, tagged as (200) and (002), reveal a predilection of

nascent In_4Te_3 samples for growth along the (040) facet, harmonizing with structural narratives in Supplementary Figure 11a–b. These revelations not only vouch for the adaptability of our fabrication approach to a gamut of domino-structured quasi-layered materials but also bolster the edifice for future explorations in this intriguing arena.

Comment 2: *At the present I would recommend publication into a more specialized journal focused on material growth. However, I would not be opposed to the Editor or other referees coming to a different conclusion.*

Author Reply: In this study, our emphasis is predominantly on presenting a comprehensive overview of a material family that resides between the realms of layered and non-layered materials, delving deep into its potential application prospects. Taking 2D QLDS-GaTe as a representative example, we extensively investigated its bonding mechanism, quantum confinement effects post two-dimensionality, second-harmonic properties, and catalytic performance, effectively inaugurating the research era for the 2D quasi-layered material family. We believe that this research framework extends our manuscript beyond mere material synthesis, directing focus to the elucidation of an entire category of materials.

In response, we are poised to further enrich our study with augmented structural characterization of the materials, their universality, and potential focal points in catalysis, paving the way for subsequent inquiries into the 2D quasi-layered material domain. Undoubtedly, the supplemental data encompassing structure, catalysis, and synthesis methodologies render our investigation more systematic and comprehensive. These revelations ensure that our contributions extend beyond the exploration of a specific material, heralding the inauguration of research into this novel material family.

Our discoveries resonate profoundly across multiple disciplines, captivating a diverse readership. The criticality of structure-dependent catalysis cannot be overstated, and there remains an evident need for augmentation. In our ensuing discussion, we underscore the pivotal structural facets and chart a future trajectory for catalysis research, establishing a solid foundation for an advanced probe into 2D QLDS materials.

Figure R3 | The EDS elemental mappings of the cross-section of the 2D QLDS-GaTe crystal.

Figure R4 | A HAADF-STEM image of the cross-section of the 2D QLDS-GaTe crystal.

Leveraging density functional theory (DFT) calculations, we delved into potential focal points in catalysis research, specifically probing whether 2D QLDS-GaTe materials of varying incline angles consistently exhibit superior catalytic activity. To ensure that this enhancement is not a peculiarity of a specific orientation, we've modelled various tilt angles in line with your recommendations, and the outcomes are presented in Figure R5–8. It's evident that while there are variations in catalytic performance with changing angles, all configurations of 2D QLDS-GaTe outperform the layered GaTe. The incorporation of these computational insights broadens our research vista, laying a robust

foundation for future exploration into the catalytic activities of 2D QLDS-GaTe structures at different orientations

Figure R5 | Calculated Gibbs free energy profiles of 2D QLDS-GaTe under the tilt angle of 61.9°.

Figure R6 | Structures of potential HER activation sites on 2D QLDS-GaTe under the tilt angle of 61.9°.

Figure R7 | Calculated Gibbs free energy profiles of 2D QLDS-GaTe under the tilt angle of 38.5°.

Figure R8 | Structures of potential HER activation sites on 2D QLDS-GaTe under the tilt angle of 38.5°.

Without question, the inclusion of supplementary data detailing structure, catalysis, and synthesis methodologies elevates the rigor and breadth of our research. Such insights guarantee that our impact is not merely confined to the scrutiny of a singular material but also signifies the dawn of inquiries into this emergent material lineage.

Revised Manuscript: (Red text: Revisions in the manuscript)

These FFT patterns align closely with the calculated simulated FFT diffraction structures (Supplementary Fig. 14), corroborating the synthesis of GaTe with a domino structure on the (-101) plane. Furthermore, using cross-sectional HADDF-STEM, we directly observed a slanted atomic arrangement in the material, offering additional validation that the material is indeed 2D QLDS-GaTe (Supplementary Fig. 15, 16).

Moreover, the reactivity of Te atoms surpasses that of layered GaTe (Supplementary Fig. 35, 36), with ΔG^* concentrated in the range of -0.5 eV to 0.5 eV, which is highly conducive to the HER process. This further validates the applicability of the Region Fold concept in domino-structure materials. Furthermore, we investigated the HER performance of Te in 2D QLDS-GaTe structures with varying tilt angles. As depicted in Supplementary Fig. 37–40, even with changes in the inclination angle, the ΔG^* corresponding to the Te atoms consistently outperforms that of layered GaTe. This discovery paves the way for future research into 2D QLDS-GaTe with different tilt angles.

Revised Supplementary Information: (Red text: Revisions in the Supplementary Information)

The EDS elemental mappings of the cross-section of the 2D QLDS-GaTe crystal. We employed W as a protective layer and conducted cross-sectional EDS elemental mapping on the material. The results revealed a uniform elemental distribution, and the interface with the Si section appeared exceptionally smooth.

A HAADF-STEM image of the cross-section of the 2D QLDS-GaTe crystal. The cross-sectional HAADF-STEM images of the material depict a remarkably smooth interface between 2D QLDS-GaTe and Si. Moreover, there's a notable correspondence with the atomic structure, further substantiating the presence of a domino structure in the derived material.

Calculated Gibbs free energy profiles of 2D QLDS-GaTe under the tilt angle of 61.9°. We constructed a model with a tilt angle of 61.9° and subsequently computed the HER performance of surface Te atoms. The results indicate that the HER activity of Te atoms at this inclination significantly surpasses that of layered GaTe.

Structures of potential HER activation sites on 2D QLDS-GaTe under the tilt angle of 61.9°. The structures below correspond to the active sites involved in the reaction, as shown in Supplementary Figure 37.

Calculated Gibbs free energy profiles of 2D QLDS-GaTe under the tilt angle of 38.5°. We constructed a model with a tilt angle of 38.5° and subsequently calculated the HER performance of surface Te atoms. The results reveal that the HER activity of Te atoms under this tilt angle markedly surpasses that of layered GaTe.

Structures of potential HER activation sites on 2D QLDS-GaTe under the tilt angle of 38.5°. The structures below correspond to the active sites involved in the reaction, as shown in Supplementary Figure 39.

To Referee #3

We gratefully acknowledge the insightful comments and constructive feedback provided by the reviewer. We have addressed each comment individually and made revisions to the manuscript accordingly. All modifications can be easily identified in the text marked in red. We trust that these revisions enhance the clarity and depth of our work, aligning it more closely with the publication standards of *Nature Communications*.

Overall remark: *This work has introduced a new concept, the 2D quasi-layered material, the interlayer of which is claimed to have synergistic blend of vdW forces and covalent bonds. The unique interlayer bonding and surface brings structural anisotropy, second harmonic generation enhancement, and hydrogen evolution reaction catalytic activity. The manuscript can be considered for publication after addressing the following issues.*

Author Reply: Firstly, we express our heartfelt gratitude to the referee for their affirmative assessment of our work. Guided by your insightful recommendations, we have meticulously refined our articulation of this emergent material paradigm to enhance its precision. In addition, we have broadened our exploration, emphasizing the unparalleled merits of our approach in fabricating 2D quasi-layered materials. By venturing further into the resilience of our growth methodology, we've successfully crafted a novel quasi-layered material embodying a domino architecture. It is undeniable that your input greatly augments the scholarly rigor of our study. We are committed to systematically addressing each of your invaluable comments in the subsequent sections.

Comment 1: *The definition of a new material by the authors seems to be mainly due to the divergence of crystal growth from the planar substrate, which may not be accurate enough. I suggest the authors to give more information on the crystal structure itself, for example, layer registry, interlayer distance, and others, before defining a new material.*

Author Reply: Thank you for your insightful feedback. In response, we have meticulously revised our articulation of this novel material paradigm, ensuring a more precise delineation. Specifically, by considering the tilt angle of crystal layers and the nuances of interlayer interactions, our characterization of the 2D quasi-layered material has been further honed and optimized. The inclination angle of the 2D quasi-layered crystal layer presents an acute angle with the plane, and the interlayer interaction is between the covalent bond and the van der Waals interaction.

Revised Manuscript: (Red text: Revisions in the manuscript)

The skewed growth mechanism engenders the formation of domino-structured material, characterized by a distinct interaction paradigm. In this model, **this exceptional material exhibits an acute angle of inclination between its layers and the horizontal plane, considered as the domino tiles. It displays special interlayer interactions that consist of a synergistic amalgamation of vdW forces and covalent bonds.** This unique amalgamation results in a force that surpasses that in layered materials, yet falls short of that in non-layered materials (Fig. 1c). Consequently, we categorize this kind of material as a quasi-layered substance.

Comment 2: *Besides of local electron microscopy characterization, it is also essential to provide macroscopic shape of the “single crystals”.*

Author Reply: Thank you for your invaluable feedback. In the manuscript, we have provided the pertinent macroscopic optical microscopy (OM) images, as illustrated in Figure R1. As evinced from the OM representation, the GaTe single crystal showcases a characteristic morphology.

Figure R1 | OM images of the 2D QLDS-GaTe single crystal.

Additionally, we have incorporated OM images of the transferred material at varying thickness levels. Recognizing that the material’s thickness is aptly represented by its contrast, we have crafted OM images for materials with different thicknesses as delineated below, which has been added to supplementary information.

Figure R2 | OM images of 2D QLDS-GaTe under diverse thicknesses.

Revised Manuscript: (Red text: Revisions in the manuscript)

Therefore, ISMG strategy with liquid metal can reduce the formation energy across an expansive chemical potential window tuned by temperatures (Supplementary Fig. 5–7) and facilitates an inclined crystal growth with a well-aligned growth orientation (Supplementary Fig. 8). Additionally, by extending the annealing duration, the concentration of Te dissolved in the liquid metal progressively increases, which in turn affects the thickness of the material (Supplementary Fig. 9, 10).

Revised Supplementary Information: (Red text: Revisions in the Supplementary Information)

OM images of 2D QLDS-GaTe under diverse thicknesses. As the annealing time increases, the amount of Te dissolved in the liquid metal also progressively rises, leading to an elevated chemical potential of Te. During the cooling process, the material tends to precipitate on the surface of the liquid metal. A higher concentration of Te accelerates this precipitation efficiency. While the material continues to exhibit a pronounced anisotropic lateral growth behavior, its vertical growth rate becomes increasingly significant. It's due to this characteristic that we can controllably synthesize 2D QLDS-GaTe of varying thicknesses. Therefore, by modulating the annealing duration, we can achieve the synthesis of a range of 2D QLDS-GaTe with diverse thicknesses.

Comment 3: *The authors have published many papers with liquid metal as nuclei to grow different materials of varying shapes. What is the advancement or the critical point to achieve this skew growth?*

Author Reply: Thank you for your invaluable suggestions. We posit that the inclined growth of the material largely hinges on the retained order within liquid-phase Ga. This characteristic is amply demonstrated in our temperature-dependent XRD results, as presented in Figure R3. As a result, the liquid metal serves as a substrate with strong interactions, potentially guiding the epitaxial growth of the material.

Figure R3 | Temperature-dependent XRD diagrams of Ga. **a**, The XRD diagrams of liquid metal under different temperature. **b**, Projected XRD results extracted from Figure R3a.

The results from DFT calculations substantiate this hypothesis aptly. With the emergence of an ordered structure, the liquid metal evolves into a substrate that exhibits strong interfacial interactions. This, in turn, facilitates significant charge interactions with the material nucleating on its surface and disrupts the conventional planar growth mode of the materials (Figure R4). Consequently, this enables the successful attainment of the tilted-growth 2D QLDS-GaTe structure.

Figure R4 | Interface interaction between Ga/2D QLDS-GaTe and Ga/layered GaTe. a, c, Differential charge density of 2D QLDS-GaTe and layered GaTe on Ga substrate, as calculated via DFT. Areas of electron accumulation and depletion are represented in yellow and cyan, respectively. b, d, Planar-averaged electron density difference plots corresponding to panels a and c.

Comment 4: *Can the method reported in this work be used to grow other materials?*

Author Reply: Your insightful feedback has been instrumental. Guided by your remarks, we delved deeper into the robustness of our growth procedure, revealing a new quasi-layered material with a domino configuration. We have successfully extended our method to the synthesis of 2D In_4Te_3 with the domino structure.

Figure R9 | Structural characterizations of 2D QLDs-In₄Te₃ samples. **a, b**, Side view and top view of 2D QLDs-In₄Te₃ structure. **c**, A low magnification TEM image of 2D QLDs-In₄Te₃ sample and EDS elemental mappings of Te and In. **d, e**, FFT patterns and HRTEM image of 2D QLDs-In₄Te₃ sample.

Figure R9a–b illustrates this unique structure, showcasing an inclined arrangement in atomic orientation. Notably, the material’s interlayer forces occupy a niche between vdW interactions and covalent bonds, directly echoing our conception of QLDs materials. Turning to the chemical characterization of the sample, we deployed EDS analysis. As detailed in Figure R9c, In₄Te₃ flakes are discerned as the primary component. Further substantiating this, EDS elemental mapping consistently portrays an even In/Te distribution across the In₄Te₃ flake, confirming material uniformity. The material’s crystalline nature is irrefutably evidenced by the singular diffraction patterns in FFT images (Figure R9d) and further corroborated by the HRTEM depiction in Figure R9e. Here, distinct interplanar spacings, denoted as (200) and (002), elucidate that the nascent In₄Te₃ samples favour growth along the (040) axis, in alignment with structural elucidation in Figure R9a–b. Such discoveries not only attest to the versatility of our synthesis technique for myriad domino-structured quasi-layered materials but also refine and fortify the foundation for subsequent investigations in this field.

Revised Manuscript: (Red text: Revisions in the manuscript)

Therefore, ISMG strategy with liquid metal can reduce the formation energy across an expansive chemical potential window tuned by temperatures (Supplementary Fig. 5–7) and facilitates an inclined crystal growth with a well-aligned growth orientation (Supplementary Fig. 8). Additionally, by extending the annealing duration, the concentration of Te dissolved in the liquid metal progressively increases, which in turn affects the thickness of the material (Supplementary Fig. 9, 10). **The ISMG strategy exhibits universality. Through this approach, we've also synthesized indium telluride materials**

with a domino structure (Supplementary Fig. 11). This discovery not only broadens the applicability of the method but also lays a foundation for unveiling more 2D QLDS materials.

Revised Supplementary Information: (Red text: Revisions in the Supplementary Information)

Structural characterizations of 2D QLDS-In₄Te₃ samples. Supplementary Figure 11a,b illuminate the distinct architecture of the material, capturing a notably tilted atomic configuration. The interlayer interactions characterizing this material intriguingly bridge the gap between vdW forces and covalent bonds, precisely resonating with the quintessential traits of QLDS materials. Transitioning to its chemical profile, we harnessed EDS techniques. Evident in Supplementary Figure 11c, the EDS elemental mapping showcases a homogenous In/Te distribution within the In₄Te₃ domain, underscoring its compositional consistency. The crystalline integrity of the material is unambiguously manifested in the singular diffraction patterns encapsulated in the FFT visuals in Supplementary Figure 11d, and this assertion gains further traction through the high-resolution TEM image depicted in Supplementary Figure 11e. Herein, prominent lattice spacings, tagged as (200) and (002), reveal a predilection of nascent In₄Te₃ samples for growth along the (040) facet, harmonizing with structural narratives in Supplementary Figure 11a–b. These revelations not only vouch for the adaptability of our fabrication approach to a gamut of domino-structured quasi-layered materials but also bolster the edifice for future explorations in this intriguing arena.

To Referee #4

We extend our gratitude for the insightful comments and constructive suggestions made by the reviewer on our manuscript. A systematic and detailed response has been prepared for each of your points, and we have prudently amended our manuscript in accordance with these valuable recommendations. Kindly refer to the marked red text for all modifications. With these revisions, we aspire to enhance the clarity and informativeness of the manuscript, aligning it more closely with the rigorous publishing standards of *Nature Communications*.

Overall remark: *The manuscript “2D Quasi-Layered Material with Domino Structure” by Lan and coauthors reports a new concept of 2D quasi-layered material, where the authors demonstrate GaTe as a representative material. The carefully tuned growth conditions enable the formation of 2D QLDS-type structure. Thanks to this unique structure, the authors found some performance improvements, such as second harmonic generation and hydrogen evolution reaction catalytic activity. The concept of structure model is new, and realization of outstanding performance would be useful for several future applications. I am positive for the publication, but before that, I would like to ask the authors to address the following questions and comments.*

Author Reply: We deeply appreciate your constructive feedback and positive stance towards our manuscript “2D Quasi-Layered Material with Domino Structure”. We are encouraged by your acknowledgment of the novelty and potential impact of our proposed 2D QLDS-type structure, particularly with GaTe as the representative material. Your insights into the significance of the performance improvements we observed, such as second harmonic generation and hydrogen evolution reaction catalytic activity, further motivate our research endeavors.

In accordance with your suggestions, we have further elaborated on the material’s definition and structural characterization, effectively enhancing the scientific rigor and precision throughout the manuscript.

Comment 1: *The structural data for 2D QLDS-GaTe is missing, such as lattice constants and fractional coordinates. As the authors imply, QLDS-GaTe is a quasi-layered structure, not the ideal layered structure. Then, what is the definition of “2D” in QLDS-GaTe? The conventional 2D model includes vacuum slab in the structure model, i.e., single carbon layer (graphene) or S-Mo-S tri-atomic layer (MoS₂). Does 2D QLDS-GaTe have vacuum slab?*

Author Reply: We are deeply grateful for your recommendations. Following your guidance, we have provided a comprehensive schematic of the material structure, as illustrated in Figure R1.

Furthermore, we enriched Figure S14 by integrating the atomistic structures utilized in the simulation, ensuring a clearer presentation of our findings.

Figure R1 | Simulated FFT results of 2D QLDS-GaTe structure with different thicknesses.

Besides, we have meticulously revised our articulation of this novel material paradigm, ensuring a more precise delineation. Specifically, by considering the tilt angle of crystal layers and the nuances of interlayer interactions, our characterization of the 2D quasi-layered material has been further honed and optimized. The inclination angle of the 2D quasi-layered crystal layer presents an acute angle with the plane, and the interlayer interaction is between the covalent bond and the vdW interaction. Additionally, the thickness of as-fabricated samples shall be lower than 20 nm.

Third, within the 2D QLDS-GaTe structure, a vacuum layer is discernible. Intriguingly, its orientation deviates at a distinct angle from the conventional 2D growth plane. Also, for all calculations related to 2D QLDS-GaTe, we introduced a vacuum layer exceeding 20 Å to isolate interactions, ensuring that the material maintained its 2D character during computations. All the structures employed

for the calculations are provided in the Supplementary Information for the convenience of researchers aiming to replicate our results. All computational outcomes incorporated this vacuum layer, guaranteeing that our models adhere to the computational standards for 2D materials. For instance, the structure utilized in our ELF calculations, as illustrated in Figure R2, incorporates a 20 Å vacuum layer to negate periodic boundary effects, ensuring that the material retains its 2D nature in the simulations.

Figure R2 | ELF diagrams of the layered GaTe and 2D QLDS-GaTe, respectively, featuring two sectional views.

Revised Manuscript: (Red text: Revisions in the manuscript)

The skewed growth mechanism engenders the formation of domino-structured material, characterized by a distinct interaction paradigm. In this model, **this exceptional material exhibits an acute angle of inclination between its layers and the horizontal plane, considered as the domino tiles. It displays special interlayer interactions that consist of a synergistic amalgamation of vdW forces and covalent bonds.** This unique amalgamation results in a force that surpasses that in layered materials, yet falls short of that in non-layered materials (Fig. 1c). Consequently, we categorize this kind of material as a quasi-layered substance.

Comment 2: *The authors calculated the phonon dispersive curves. I'm wondering whether 2D QLDS-GaTe is a new crystal polymorph of GaTe or it's same as already known structure, but just preferred crystal orientation is applied on the substrate. The vibration property of GaTe was reported, for example, Scientific Reports 11, 21202 (2021). Is this reported structure same as the model mentioned in this manuscript?*

Author Reply: We appreciate your discerning remarks. In our investigation, the 2D QLDS-GaTe we

synthesized represents a fresh growth orientation built upon established crystallographic foundations. While prior studies have extensively elucidated the crystalline structure of GaTe (*Sci. Rep.* **2021**, *11*, 21202), they predominantly yielded 2D layered configurations. Such architectures arise from the inherent tendency of atoms to assume densely packed configurations. This spatial arrangement tends to diminish entropy, thereby enhancing the material's structural robustness.

In contrast, our fabricated structure deviates from the traditional 2D layered construct, showcasing an inclined growth orientation. This distinctive configuration undeniably gives rise to unique vibrational modes, divergent from those in layered GaTe. As such, our findings offer invaluable insights, particularly for acoustic studies of the material.

Comment 3: *The low-magnification TEM pictures in Fig.2 are hard to understand. The authors should indicate the schematic sample structure. Moreover, the width of the sample, where Ga and Te were detected, is as broad as 1 μm . This looks very thick in terms of 2D layer.*

Author Reply: Thank you for your constructive feedback. The side view of the structure has been displayed on Figure R1 and revised in Figure S14. The low-magnification TEM images we presented aim to showcase the morphology of our synthesized material, emphasizing its elongated, thin-strip form. Contrary to potential misinterpretation, this content does not represent cross-sectional EDS mappings but results observed from a planar perspective. The width of the sample represents the lateral dimension rather than the thickness. To directly prove that the sample we synthesized is the 2D material, we have provided AFM results of the sample. Figure R3 (also seen in Figure S13 in the revised manuscript) shows that the as-synthesized sample has an ultra-thin thickness.

Figure R3 | AFM image of the 2D QLDS-GaTe crystal.

Meanwhile, the low-magnification TEM images we presented aim to showcase the morphology of our synthesized material, emphasizing its elongated, thin-strip form. Contrary to potential

misinterpretation, this content does not represent cross-sectional EDS mappings but results observed from a planar perspective. To circumvent any confusion, we have incorporated additional cross-sectional EDS mappings and adjusted the corresponding figure captions in the manuscript accordingly.

Figure R3 | The EDS elemental mappings of the cross-section of the 2D QLDS-GaTe crystal.

Revised Manuscript: (Red text: Revisions in the manuscript)

i, Low-magnification STEM image accompanied by corresponding EDS elemental mapping of Ga, Te, and the overlay of Ga and Te elements, **observing from the planar perspective.**

Revised Supplementary Information: (Red text: Revisions in the Supplementary Information)

The EDS elemental mappings of the cross-section of the 2D QLDS-GaTe crystal. We employed W as a protective layer and conducted cross-sectional EDS elemental mapping on the material. The results revealed a uniform elemental distribution, and the interface with the Si section appeared exceptionally smooth.

Comment 4: *Fig3e has no information on thickness. Please add legends.*

Author Reply: Thank you for your suggestion. We have supplemented the thickness information corresponding to Figure 3e, and the updated representation can be seen in Figure R4.

Figure R4 | SHG spectra corresponding to each thickness of the sample.

Comment 5: *I'm curious to know the thickness dependence of FWHM of SHG spectra. Is it constant, or is there any dependence similar to SHG intensity?*

Author Reply: Thank you for your invaluable input. We have plotted the relationship between FWHM and thickness, as shown in Figure R5. There appears to be no discernible thickness dependency; the overall trend remains relatively consistent, fluctuating only within a narrow range.

Figure R5 | Thickness-dependent FWHM relationship diagram.

Comment 6: *The improvement of the catalytic reactions is explained due to the presence of the edge states. This may be related to the non-bonded electrons at the surface atoms, which are absent in the conventional surface terminated 2D materials. If this is the case, the random oriented polycrystalline GaTe film could show a similar performance as it also possesses unsaturated edge states. Why was this enhancement observed only for the 2D QLDS-GaTe?*

Author Reply: Thank you for your constructive feedback. In polycrystalline materials, while there exists an array of defect states, it is the prevalent grain boundaries and inherent defects that critically impede their conductivity, leading to a marked suppression of catalytic activity (*Adv. Sci.* **2020**, *7*, 2002172).

With respect to the synthesized 2D QLDS-GaTe, it exhibits superior crystallinity compared to polycrystalline materials. Thus, considering the polycrystalline feature, significant electron scattering occurs at the grain boundaries, leading to a reduction in the material's conductivity. This phenomenon impedes the flow of charge carriers, making the material less efficient in terms of electrical performance compared to its single-crystalline counterparts (*Science* **2018**, *362*, 1021). However, conductivity plays a pivotal role in electrocatalytic reactions, which directly result in the lower catalytic performance for polycrystalline samples.

Therefore, we probed the conductive nature of 2D QLDS-GaTe material, as depicted in Figure R6. The I_{ds} - V_{ds} characteristic curves validate the successful formation of ohmic contact between the material and the electrodes. Simultaneously, as the gate voltage is modulated, a substantial shift in conductivity is discerned, underscoring the fact that 2D QLDS-GaTe maintains its inherent semiconducting attributes. Such traits position this material as a promising candidate for integration into field-effect transistors and photodetectors. Furthermore, the potential for gate voltage-regulated catalytic processes is another intriguing avenue that warrants deeper exploration.

Figure R6 | I_{ds} - V_{ds} characteristic curves with different gate voltages.

For polycrystalline materials, achieving proper conduction is inherently challenging. Although such materials possess an abundance of grain boundaries and defects, their inferior conductivity typically places them at a disadvantage in catalytic applications. This compromised conductivity often overshadows the potential benefits these defects and grain boundaries might offer in specific reactions or processes.

REVIEWER COMMENTS

Reviewer #1 (Remarks to the Author):

I am glad to see the authors have provided some evidence to demonstrate the domino structure. However, the mechanistic aspects are still unsatisfactory. The following two major issues must be addressed before it might be recommended for publication:

Generally, the raise of a new 'concept' is a very serious action. In this work, the definition of the so-called 'Region Fold concept' is highly ambiguous and unnecessary.

First, the authors claim that the 'Region Fold concept' pertains to a phenomenon observed in the past research in Adv. Mater. 2016, 28, 6197; Nat. Commun. 2019, 10, 1217; and Nat. Commun. 2022, 13, 2193. However, no related information was found in above papers.

Second, according to the elucidation in response letter, the 'Region Fold concept' just refers to the fact that edge sites show higher activity than basal plains, which has been recognised for a long time by catalysis community. Therefore, the raise of such 'Region Fold concept' just makes no sense.

Besides, the response to Comment 7 is unconvincing. The statement that 'while electron acceptors are pivotal in driving the transformation of the transformation of H^+ to H^* , electron donors chiefly mediate the progression from H^* to H_2 ' is highly unlogic.

First, the transformation of H^+ to H^* requires the participation of electron donors, not electron acceptors.

Second, the authors claim that a recent study (Nat. Commun. 2022, 13, 2193) illustrates the same mechanism. However, the case in above paper is totally different from that in this work.

Third, the authors said 'both Ga and select Te atoms feature orbitals at the Fermi level, ready to accept electrons from adsorbed species' in response letter, which is not the case of HER. HER occurs at the cathode surface, where H^+ accepts electrons from the electrode. I donot reckon a 'donor and acceptor' theory is applicable in acidic HER.

Basically, the structure is fancy and the activity is seemingly good. But the mechanistic elucidation is rather indefensible. The authors should focus more on the intrinsic activity rather than a questionable theory.

Reviewer #2 (Remarks to the Author):

Given the more detailed revisions based on suggestions from the other referees ... I would leave it to their discretions whether or not the authors have met their expectations.

I feel that this work should be considered by a more specialized journal. I am not sure it meets the novelty expected for Nature Comms.

Reviewer #3 (Remarks to the Author):

My concerns and questions have been carefully addressed in this version. I think this manuscript can be accepted for publication now.

Reviewer #4 (Remarks to the Author):

The authors addressed all questions and comments. I believe that the manuscript should be published as is.

RESPONSE TO REVIEWERS' COMMENTS

To Referee #1

We sincerely thank your insightful feedback and constructive suggestions on our manuscript. We have diligently addressed each comment and have thoroughly revised the manuscript in accordance with your recommendations. Our revisions are clearly highlighted in red.

We hope that this comprehensively updated version provides clearer insights and holds greater informative value. We believe it now aligns with the stringent criteria set by *Nature Communications* for publication.

Overall remark: *I am glad to see the authors have provided some evidence to demonstrate the domino structure. However, the mechanistic aspects are still unsatisfactory, The following two major issues must be addressed before it might be recommended for publication:*

Author reply: We are deeply grateful for your insightful advice and assistance. We recognize that our performance in catalysis and mechanistic studies was not up to the mark, and we sincerely apologize for this shortcoming. In light of your and the editor's feedback, we have chosen to exclude the section related to HER research. Your guidance has been invaluable, and we express our profound appreciation.

Comment 1: *First, the authors claim that the Region Fold concept' pertains to a phenomenon observed in the past research in Adv. Mater. 2016, 28, 6197; Nat. Commun. 2019, 10, 1217; and Nat. Commun. 2022, 13, 2193. However, no related information was found in above papers.*

Author reply: We wish to extend our sincere gratitude to the reviewer for their insightful feedback and constructive suggestions regarding our manuscript. We have diligently addressed your comments in a systematic and comprehensive manner.

The 'Region Fold concept' is an interpretation we derived from our analysis of existing literature, rather than a direct report from the literature itself.

The referenced literature delves into the catalytic activity of the conventional layered material, MoS₂, highlighting its in-plane catalytic properties. A comparison between the catalytic activity of pristine MoS₂ described in the literature and the edge activity measured in our MoS₂ samples reveals a pronounced superiority of the edge sites. Thus, identifying a growth orientation that maximally exposes the edge area of the material would be of significant merit.

In refining our work with insights from existing literature, we distilled what we termed the 'Region Fold concept'. However, upon your astute observation, we recognize its lack of rigor and have decided

to revise and omit it from the manuscript.

Comment 2: *Second, according to the elucidation in response letter, the “Region Fold concept” just refers to the fact that edge sites show high electivity than basal plains, which has been recognised for a longtime by catalysis community. Therefore, the raise of such “Region Fold concept” just makes no sense.*

Author reply: We deeply appreciate your invaluable insights regarding this theory. Our introduction of the 'Region Fold concept' was primarily to elucidate the superior HER performance stemming from this material's novel structure. It is a widely accepted observation within the research community that, in layered materials, edge sites often exhibit greater activity than the basal plane (Nat. Nanotechnol. 2016, 11, 218–230).

Heeding your advice, we have opted to omit this description to prevent potential ambiguities and misconceptions.

Comment 3: *Besides, the response to Comment 7 is unconvincing. The statement that while electron acceptors are pivotal in driving the transformation of the transformation of H^+ to H^* electron donors chiefly mediate the progression from H^* to H_2 is highly unlogic. First, the transformation of H^+ to H^* requires the participation of electron donors, not electron acceptors.*

Author reply: We extend our sincere gratitude for pointing out this oversight. As you correctly noted, the conversion of H^+ to H^* necessitates the participation of an electron donor, rather than an electron acceptor. We have rectified this in the main text and excised the ambiguous segments. Your insightful feedback is much appreciated.

Comment 4: *Second, the authors claim that a recent study (Nat. Commun2022, 13, 2193) illustrates the same mechanism. However, the case in above paper is totally different from that in this work.*

Author reply: Thank you for your invaluable suggestions. Our citation of that particular literature was intended to draw a comparison between the in-plane catalytic activity of traditional layered materials and the edge catalytic activity we observed. This led us to conclude that edge activity significantly surpasses in-plane activity. Consequently, maximizing the exposure of material edges can notably enhance catalytic performance.

This aligns well with our identified domino structure, which achieves maximum edge exposure, exhibiting outstanding catalytic activity and unique electronic states. However, to prevent potential ambiguities, we have decided to remove this content. We once again appreciate your insights.

Comment 5: *Third, the authors said both Ga and select Te atoms feature orbitals at the Fermi level, ready to accept electrons from adsorbed species in response letter. which is not the case of HER. HER occurs at the cathode surface, where H^+ accepts electrons from the electrode. I do not reckon a 'donor and acceptor' theory is applicable in acidic HER.*

Author reply: We are deeply grateful for your insightful feedback. We appreciate your identification of the error in our manuscript regarding the HER process, where H^+ indeed accepts electrons from the electrode. We have rectified this segment and removed the erroneous content. Your guidance is invaluable.

Comment 6: *Basically the structure is fancy and the activity is seemingly good. But the mechanistic elucidation is rather indefensible. The authors should focus more on the intrinsic activity rather than a questionable theory.*

Author reply: We extend our profound gratitude for your constructive feedback. We are also thankful for your recognition of the structures and concepts presented in our manuscript. We apologize for any inadequacies in our theoretical analysis.

Following your and the editor's joint recommendations, we have excised the contentious sections, including those on HER theory, to ensure the rigor of our content. Your guidance has been instrumental, and we truly appreciate it

To Referee #2

We are grateful for the thoughtful comments and suggestions provided by the reviewer. Each comment has been addressed comprehensively, and we have undertaken revisions in accordance with your recommendations. It is our aspiration that these modifications have enhanced the clarity and depth of the manuscript, aligning it with the publication standards of *Nature Communications*.

Overall remark:

Given the more detailed revisions based on suggestions from the other referees ...I would leave it to their discretions whether or not the authors have met their expectations.

I feel that this work should be considered by a more specialized journal. I am not sure it meets the novelty expected for Nature Comms.

Author Reply: We deeply appreciate your insights, and earnestly hope for your endorsement as well.

The 2D quasi-layered material investigated in our manuscript represents a unique category of materials, bridging the gap between 2D layered and non-layered entities. This discovery effectively fills the void between these classifications, offering a fresh perspective in the structural modification studies of 2D materials. Additionally, this material exhibits exceptional optical properties and distinctive surface activity, laying the groundwork for future applications in optoelectronics, catalysis, and beyond.

We are convinced of the pivotal significance of introducing this concept, a sentiment we believe is shared by the editor and fellow reviewers. Your affirmation holds particular weight for us, and we sincerely hope to gain your endorsement.

To Referee #3

Overall remark:

My concerns and questions have been carefully addressed in this version. I think this manuscript can be accepted for publication now.

Author Reply: We deeply appreciate your thorough review and the time you dedicated to assessing our manuscript. Your constructive feedback was instrumental in refining our work. We are sincerely grateful for your positive evaluation and the affirmation to move forward with publication. Your insights and expertise have greatly benefited our study. Thank you once again for your invaluable contribution to this manuscript.

To Referee #4

Overall remark:

The authors addressed all questions and comments. I believe that the manuscript should be published as is.

Author Reply: Thank you for your meticulous evaluation and thoughtful feedback on our manuscript. We are immensely grateful for your positive assessment and your belief in the merit of our work. Your comments were instrumental in refining our research presentation, and we are heartened by your endorsement for publication. We deeply value your contributions to the quality and integrity of our work.

REVIEWERS' COMMENTS

Reviewer #1 (Remarks to the Author):

I think the current version is suitable for publication. I have no further comments.

RESPONSE TO REVIEWERS' COMMENTS

To Referee #1

Overall remark: *I think the current version is suitable for publication. I have no further comments.*

Author Reply: We want to express our heartfelt gratitude for your comprehensive review and the dedicated time you devoted to evaluating our manuscript. Your constructive feedback has played a pivotal role in enhancing the quality of our work. We genuinely appreciate your positive assessment and your encouragement to proceed with the publication. Your valuable insights and expert input have significantly enriched our study. Once again, we extend our sincerest thanks for your invaluable contribution to this manuscript.